# Recent Advances in the Efficient Synthesis of Useful Amines from Biomass-Based Furan Compounds and Their Derivatives over Heterogeneous Catalysts

**Jia Zhang** [1,2]**, Jian Yang** [1]**, Xuemei Li** [1]**, Hailong Liu** [1,*]**, Xiaolan Yao** [1]**, Chungu Xia** [1] **and Zhiwei Huang** [1,*]

[1]   State Key Laboratory for Oxo Synthesis and Selective Oxidation, Lanzhou Institute of Chemical Physics (LICP), Chinese Academy of Sciences, Lanzhou 730000, China
[2]   University of Chinese Academy of Sciences, Beijing 100049, China
*   Correspondence: hlliu@licp.cas.cn (H.L.); zwhuang@licp.cas.cn (Z.H.); Tel.: +86-93-1496-8129 (Z.H.)

**Abstract:** Bio-based furanic oxygenates represent a well-known class of lignocellulosic biomass-derived platform molecules. In the presence of $H_2$ and different nitrogen sources, these versatile building blocks can be transformed into valuable amine compounds via reductive amination or hydrogen-borrowing amination mechanisms, yet they still face many challenges due to the co-existence of many side-reactions, such as direct hydrogenation, polymerization and cyclization. Hence, catalysts with specific structures and functions are required to achieve satisfactory yields of target amines. In recent years, heterogeneous catalytic synthesis of amines from bio-based furanic oxygenates has received extensive attention. In this review, we summarize and discuss the recent significant progress in the generation of useful amines from bio-based furanic oxygenates with $H_2$ and different nitrogen sources over heterogeneous catalysts, according to various raw materials and reaction pathways. The key factors affecting catalytic performances, such as active metals, supports, promoters, reaction solvents and conditions, as well as the possible reaction routes and catalytic reaction mechanisms are studied and discussed in depth. Special attention is paid to the structure–activity relationship, which would be helpful for the development of more efficient and stable heterogeneous catalysts. Moreover, the future research direction and development trend of the efficient synthesis for bio-based amines are prospected.

**Keywords:** bio-based furanic oxygenate; amine; reductive amination; hydrogen-borrowing amination; heterogeneous catalysis; structure–activity relationship





## 1. Introduction

An important challenge facing mankind in the 21st century is how to meet the growing energy demand while reducing greenhouse gas emissions [1,2]. Under the dual pressure of resources and environment, the exploitation of clean and renewable resources is greatly promoted. Biomass is the most abundant renewable carbon resource in nature, with an annual natural output of about 170 billion tons [3,4]. It is considered as a green chemical raw material due to its special advantages in terms of sustainability. For example, by 2030, the EU aims to increase the total use of bio-based chemicals and materials to 25% and to reduce greenhouse gas emissions by 40% [5]. Therefore, many governments and research institutions are paying great attention to the research on the replacement of traditional fossil resources by biomass.

Biomass-derived platform compounds have attracted great attention in reducing the dependence on fossil resources, as they are regarded as the bridges between biomass resources and the chemical industry [6–8]. Non-edible lignocellulose is a major component of biomass resources, which consists of polysaccharides (30–50 wt% cellulose and 20–40 wt% hemicellulose) and phenolic compounds (10–20 wt% lignin) [9,10]. Based on chemical or

biological catalytic technologies, polysaccharides can be readily converted into biomass-based oxygenated platform compounds, such as furfural (FF), 5-hydroxymethylfurfural (HMF) and their derivatives [6,11] (Figure 1). The use of these low-cost and readily available furan-based oxygenates for the synthesis of high-value-added chemicals, especially various useful amines, has become a research hotspot in biorefinery in the last decade [12–15].

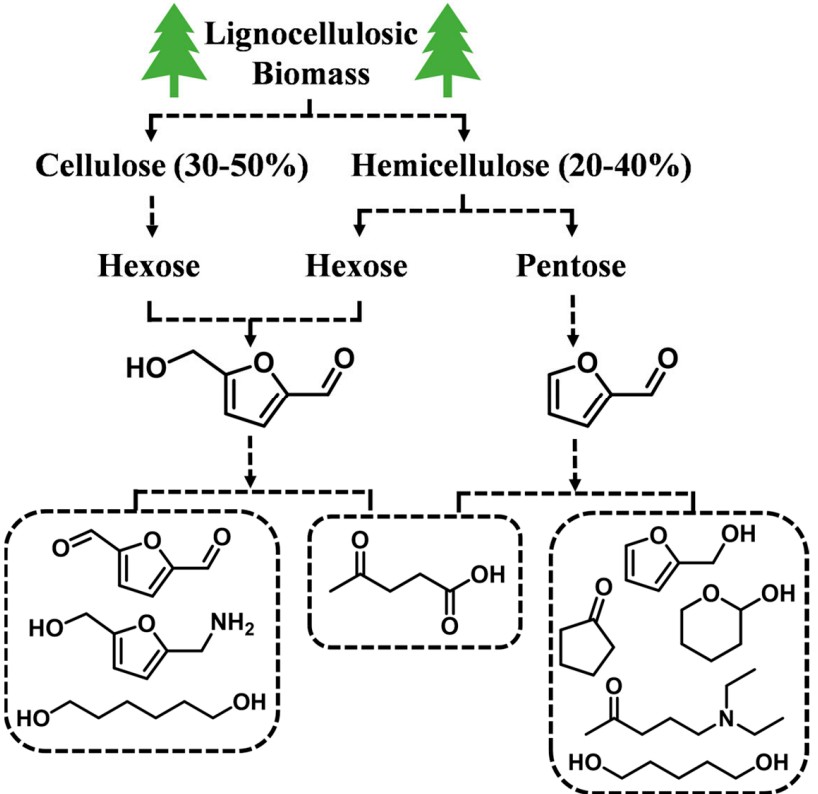

**Figure 1.** Schematic representation of bio-based furanic oxygenates.

Amines, an important group of stock molecules in the chemical and biological sciences, are widely employed as synthetic raw materials and essential intermediates in a variety of industries, including pharmaceuticals, insecticides, polymers, organic pigments, food additives and surfactants [14,16–18]. For example, in the field of drug synthesis, more than 80% of the 200 top-selling drugs in 2018 contained amines [13,19]. In the field of polymer synthesis, the annual world production of polyamide has reached a market scale of 8232 kilotons in 2018 and will reach more than 10,900 kilotons by 2025 [20]. Traditionally, amine synthesis routes are the direct amination of alkyl halides or epoxides, and the hydrogenation of nitriles or amides [20–22]. These processes not only depend on fossil materials, but also suffer from being high-energy intensive and/or producing heavy pollution. On the other hand, the amination of carbonyl and alcohol-based oxygenates derived from biomass undoubtedly offers a cleaner and more atom-efficient route with water as the only by-product, which complies with the 12 green chemistry principles [7,21]. In recent years, significant progress has been made in the amination of bio-based furanic oxygenates into amine products [23–28]. In this context, recent advances in the synthesis of bio-based furanic amines are highly worth reviewing in order to provide a reference for the future development of the sustainable pharmaceutical and polymer industries.

Herein, we focus on the latest progress of efficient heterogeneous catalytic systems and strategies for the amination of bio-based furanic oxygenates to useful amines, including furfurylamine, aminoalcohols, diamines, N-heterocyclic amines and so on, in the presence of molecular $H_2$ as a green reductant. Different raw materials and reaction types are classified and summarized (Figure 2). A comprehensive evaluation of

the factors affecting catalytic performance, reaction conditions and mechanisms, as well as structure–activity relationships is presented to lay the experimental and theoretical foundations for the development of novel and stable catalysts that can be used for the amination of not only bio-based furanic oxygenates, but also other special oxygenates with multifunctional groups.

**Figure 2.** Valorization strategy for bio-based furanic oxygenates by reductive amination and hydrogen-borrowing amination.

## 2. Reductive Amination of Bio-Based Furanic Aldehydes and Ketones

The reductive amination of aldehydes and ketones is an impressive approach for the synthesis of useful amines with mild reaction conditions and water as the main by-product. It is widely accepted that the reductive amination reaction involves the condensation of the carbonyl group with ammonia to form an imine intermediate, followed by the hydrogenation of the imine intermediate to yield the desired amine product (Scheme 1) [21], where the hydrogenation of imine in the second step is somewhat more difficult, likely being the rate-determining step. Achieving a highly selective synthesis of the desired amine, especially primary amine, is a difficult task due to the presence of common side-reactions, such as direct reduction of the carbonyl and overalkylation of the target amine [13,29]. Therefore, it is promising to select a catalytic system that favors the hydrogenation of imine over the hydrogenation of the carbonyl substrate and overalkylation of the target amine [30–34].

### 2.1. Reductive Amination of FF

FF is one of the most prevalent industrial chemicals derived from lignocellulosic biomass, with an annual production volume of more than 200,000 tons. It can be facilely obtained by the dehydration of pentose over acidic catalysts and produced industrially on a large scale from corncobs, bagasse, wheat straw and other agroforestry wastes by hydrolytic refining [35,36]. As a promising platform compound, FF can be transformed into a wide range of value-added chemicals and biofuels [37,38]. Among the different routes for upgrading, the reductive amination of FF to furfurylamine (FAM) and its derivatives has attracted much attention in recent years due to the broad application of FAM and its derivatives in the manufacture of pharmaceuticals, pesticides and synthetic resins [39–41]. In this section, the reductive amination of FF will be discussed on the basis of different nitrogen sources.

### 2.1.1. Reductive Amination of FF with $NH_3$

The reductive amination of FF with $NH_3$ is a typical method for the synthesis of FAM. A plausible reaction route for the synthesis of FAM and its by-products is as follows (Scheme 2) [23,28,42]: FF (1a) is reversibly condensed with excess $NH_3$ to form an unstable

intermediate of primary imine (2a), and then 2a is hydrogenated to obtain the target FAM (3a) over an active catalyst. Meanwhile, at least three side-reactions compete with the main reaction, such as the direct hydrogenation of FF to furfuryl alcohol (5a), the trimerization and cyclization of imine (2a) to form 10a, and the condensation of 1a with 3a to form a stable intermediate of secondary imine (7a), which can be further hydrogenated to secondary amine (8a). Therefore, the selection of catalysts with special structures and functions plays an important role in achieving an efficient synthesis of FAM by inhibiting the occurrence of side-reactions.

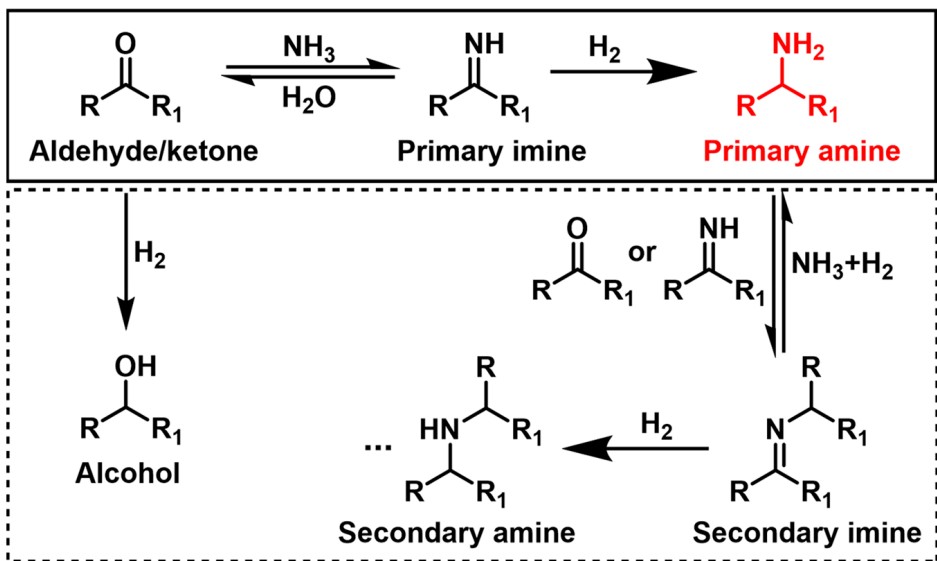

**Scheme 1.** Reaction pathway for the reductive amination of aldehydes and ketones. Adapted with permission from ref. [30]. Copyright 2020 American Chemical Society.

**Scheme 2.** Reaction pathway for the reductive amination of FF with NH₃.

In searching for efficient catalysts for the reductive amination of FF to FAM, Hara and coworkers [43] investigated a wide range of metals (Ru, Rh, Pd, Pt, Ag, Ni and Cu) and oxide supports (Nb$_2$O$_5$, SiO$_2$, TiO$_2$, Al$_2$O$_3$, ZrO$_2$, MgO and active carbon). Among them, Ru/Nb$_2$O$_5$ showed the highest FAM yield of 99% under the reaction conditions of 90 °C, 0.1 MPa NH$_3$ and 4 MPa H$_2$. The low electron density of Ru on the positively charged Nb$_2$O$_5$ surface greatly contributed to the highly selective reductive amination of FF to FAM. The effect of the type and amount of acid was also examined by systematically adjusting the reduction temperature of the Ru/Nb$_2$O$_5$·$n$H$_2$O catalyst [44]. Ru/Nb$_2$O$_5$·$n$H$_2$O-300 with a high density of acidic sites had the highest catalytic activity, which could facilitate the attack of Ru hydride species by activating imines, thus accelerating the hydrogenation of imines. A specific flat-shaped Ru nanoparticles (Ru-NPs) catalyst with higher activity

was synthesized by the same group [45]. The conversion frequency (TOF) of the Ru-NPs catalyst was as high as 1850 h$^{-1}$, which was almost six times that of the Ru/Nb$_2$O$_5$ catalyst (TOF = 310 h$^{-1}$). The high activity of the catalyst was attributed to the exposure of a large number of electron-deficient Ru as active sites on the (111) surface of flat-shaped fcc Ru nanoparticles. The Ru-NPs catalyst showed excellent durability in long-term cycle tests.

Most current processes for producing primary amine by reductive amination are carried out with extremely excessive NH$_3$, which raise the cost of recovery and have negative environmental effects. Xu et al. [46] designed an efficient anisotropic layered boron nitride-supported ruthenium catalyst (Ru/BN-e), which successfully achieved the quantitative conversion of FF to FAM in the presence of a relatively low amount of NH$_3$ (2 eq) aqueous solution. The remarkable catalytic activity of Ru/BN-e was specifically related to the enhanced interface electronic effect between Ru and the edge surface of boron nitride, which consequently enhanced the hydrogenation activity.

Bimetallic catalysts can modify the surface characteristics of the catalyst through the interaction of different metals to considerably improve the catalytic performance [47,48]. Ru-based bimetallic catalysts supported on activated carbon (RuM/AC) were reported by Dai et al. [39] for the reductive amination of FF to FAM, and a 92% yield of FAM was obtained over the RuCo/AC catalyst under the reaction conditions of 80 °C and 2 MPa H$_2$, using water as a green solvent. The good catalytic performance of RuCo/AC was ascribed to the synergistic effect between the RuCo alloy and the activated carbon, which improved its acidity and hydrogenation ability. The catalyst could be recycled five times without significant deactivation.

Single-atom catalysts (SACs), unlike conventional supported catalysts, have emerged as a new field of heterogeneous catalysis due to their well-defined active site and maximum metal atom utilization [49,50]. The group of Zhang and Wang [28] fabricated highly active, selective and robust Ru-SACs supported on N-doped carbons (Ru$_1$/NC-900–800NH$_3$) by pyrolyzing Ru(acac)$_3$ and N/C precursors at 900 °C in N$_2$ and then treating the mixture at 800 °C in NH$_3$ (Figure 3). The Ru$_1$/NC catalyst can afford a good yield (97%) toward FAM in the reductive amination of FF, owing to its moderate hydrogen activation capacity. The catalyst also showed outstanding stability during reuse tests and universality to a wide range of biomass-derived aldehyde/ketone. More intriguingly, Ru$_1$/NC SAC displayed superior sulfur and CO resistance compared with traditional Ru-based nano-catalysts.

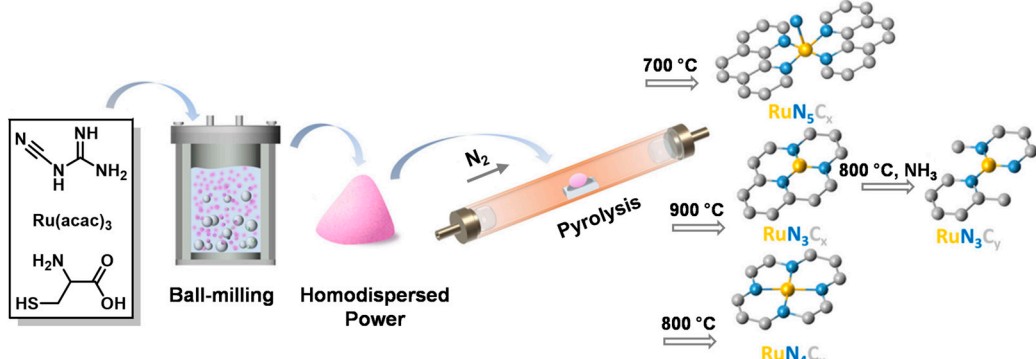

**Figure 3.** Schematic illustration for the synthesis procedure of Ru$_1$/NC-T catalysts. Adapted with permission from ref. [28]. Copyright 2021 The Author(s).

It is a great challenge to efficiently synthesize primary amines via ambient temperature reductive amination of carbonyl compounds. Recently, using a titanium phosphate (TiP)-supported Ru catalyst (Ru/TiP-100) reduced at 100 °C, Han and coworkers [33] carried out the reductive amination of FF and obtained a 91% yield of FAM at room temperature. The good catalytic performance of the Ru/TiP-100 catalyst originated from the relatively high acidity and the suitable electron density, provided by the combination of TiP and Ru/RuO$_2$ with a suitable proportion of Ru$^0$ (52%). Detailed studies showed that the suitable electron

density of Ru species could balance the activation of $H_2$ in the hydrogenation step and the desorption of intermediates (primary and secondary imines). Meanwhile, the relatively high acidity of Ru/TiP-100 could facilitate the conversion of the C=N groups in the in situ-generated primary and secondary imines to the desired primary amines.

Despite Ru-based catalysts showing good catalytic performance in the synthesis of FAM, their expensive production costs and lack of availability retard them from large-scale industrial applications. Therefore, it is important to develop cheap and easily available non-noble metal catalysts for the reductive amination of FF. Xu et al. [51] studied the reductive amination of FF on a commercially available Raney Co catalyst, achieving an FAM yield up to 99%. The excellent catalytic performance of Raney Co was attributed to its high efficiency in the ammonia-assisted hydrogenolysis of the secondary imines and low activity in the further hydrogenation of the secondary imines or the direct hydrogenation of the carbonyl and the furan ring in FF.

Recently, Ma and coworkers [52] successfully synthesized a uniform Co nanoparticles catalyst (Co@C-600-EtOH) enclosed in a multilayer graphene structure using cobalt acetate as the metal precursor. Such a catalyst achieved an 87% FAM yield under the reaction conditions of 90 °C, 7 M $NH_3$ in MeOH and 2 MPa $H_2$. The inserted Co species was found to function as electrophilic sites or Lewis acidic sites to promote imine formation and hydrogenation. Subsequently, Lingaiah et al. [53] prepared Co nanoparticles embedded in a N-doped carbon matrix by pyrolyzing ZIF-67 in a $N_2$ atmosphere at different temperatures (400–700 °C). The Co/NC-700 catalyst could achieve a rather high FAM yield of 99% under the reaction conditions of 120 °C and 2 MPa $H_2$. They found that the active center of the catalyst was the metallic Co nanoparticles anchored by coordination with the N species in the graphitic layers. Meanwhile, the large surface area, total pore volume, easy hydrogen desorption and surface defect sites together contributed to good catalytic activity, selectivity and stability of the catalyst. The direct conversion of biomass-derived xylose into FAM through a one-pot two-step process was also accomplished, with an FAM yield of 69%.

In searching for facilely synthesized non-noble Ni-based catalysts, Ding et al. [54] prepared several nano-Ni catalysts loaded on commercially available oxide supports using a simple wet impregnation method. Among them, $10Ni/Al_2O_3$ was found to present superior activity and selectivity for FAM with a high yield of 92%. The synergistic effect of medium acidic sites and strong metal-support interaction was disclosed to be responsible for the remarkable performance of the catalyst. Recently, using a nano-$Ni_1Al$ catalyst with a Ni:Al ratio of 1:1 prepared by coprecipitation, Zhang et al. [55] attained a 91% yield of FAM in the reductive amination of FF over the $Ni-Al_2O_3$ nano-catalysts. The synergistic effect between the well-dispersed active $Ni^0$ nanoparticles and the abundant adjacent surface Lewis acidic sites accounts for the better catalytic performance of the catalyst with a Ni/Al ratio of 1. More recently, up to a 98% yield of FAM was obtained over a $Ni/SiO_2$ catalyst with small Ni sizes (~3 nm) prepared by deposition–precipitation in the reductive amination of FF [56]. A similar cooperative catalysis mechanism of the Lewis acidic sites and the small nickel particles of the catalyst was proposed for the efficient synthesis of FAM.

N-doped carbon materials were also applied for the fabrication of supported non-noble Ni-based catalysts. For instance, Song et al. [57] created an effective N-doped porous carbon-supported Ni catalyst (Ni/pNC) with a uniform pore structure utilizing a template-assisted pyrolysis and impregnation approach. The Ni/pNC catalyst showed a >99% yield of FAM. The formation of $Ni–N_x$ sites and the electronic interaction between the Ni and N species were found to facilitate the reductive amination of the C=O bonds and significantly decrease the activation energy for the conversion of trimers and secondary imines to the target amines. Additionally, the Ni/pNC catalyst demonstrated good durability and broad application in the reductive amination of various carbonyl compounds with high primary amine yields.

The development of heterogeneous non-noble metal catalysts for the efficient reductive amination of FF under quite mild conditions is still challenging. The Hara group [58] used

a stable Ni@DS catalyst, which was prepared by low-temperature pyrolysis of polydentate oxygen-coordinated chelating ligands and mesoporous dendritic silica supports, for the reductive amination of FF to produce FAM with a yield of 89% under rather mild conditions of 50 °C and 0.9 MPa $H_2$. This catalyst showed high activity in both water (polar and protic) and toluene (apolar) solvents, which might be attributed to the synergistic effect between small metals and their metal oxide nanoparticles uniformly fixed on silica.

### 2.1.2. Reductive Amination of FF with Aniline

One example of the heterogeneous catalytic reductive amination of FF with aniline at room temperature was reported by Martínez and coworkers [59]. They applied sulfonic acid functionalized silica as a support to fabricate a bifunctional $Ir/SiO_2-SO_3H$ catalyst. A 21% yield of the target secondary amine (*N*-(furan-2-ylmethyl)aniline) was obtained at room temperature using ethyl acetate as the solvent, which might be associated with the fact that use of ethyl acetate as a solvent inhibited the formation of a tertiary amine by-product. The synergistic effect of the metal center and acidic sites exerted a vital role in the reductive amination of FF with aniline, in which acidic sites facilitated both the formation of imine and its subsequent hydrogenation to the desired amine. However, according to the results, the $Ir/SiO_2-SO_3H$ catalyst is not active enough to be used in the reductive amination of FF with aniline.

### 2.1.3. Reductive Amination of FF with $HCOONH_4$

Tertiary amines, mainly produced synthetically from fossil resources using a multi-step process, are essential substances in the chemical industry. Lin et al. [60] reported the first example of the continuous reductive amination of FF with $HCOONH_4$ to synthesize a biomass-based tertiary amine of *tris*(2-furanylmethyl)amine. Over an efficient $Rh_2P/NC$ catalyst, up to a 92% yield of *tris*(2-furanylmethyl)amine was achieved under mild reaction conditions of 60 °C and 3 MPa $H_2$ in an ethyl acetate solvent. The good performance of the catalyst was ascribed to the efficient electron-transfer from the P atoms (on the P-terminated $Rh_2P$) to their bonded and dissociated H atoms, which resulted in the partial filling of the antibonding orbitals of the H atoms, favoring the adsorption and activation of $H_2$ on the surface of $Rh_2P$. The $Rh_2P/NC$ catalyst showed good durability in six continuous cycle tests without a noticeable decrease in activity.

An overview of the reductive amination of FF with different nitrogen sources is summarized in Table 1. Most of the studies have focused on the reductive amination of FF with $NH_3$. The properties of the metal have a significant influence on the selectivity of FAM, where metals with too high or too low hydrogenation activity are not favorable for the formation of FAM, generally improving in the order of Pt < Pd < Rh < Ru < Ni < Co [31,56,61]. Noble Ru and non-noble Ni and Co-based catalysts with moderate hydrogenation capabilities have been extensively investigated and showed remarkable performance, achieving near quantitative yields of the target FAM. Even Ru-based catalysts loaded on supports with specific properties such as titanium phosphate were capable of converting FF to FAM at ambient temperature. In general, the rational design strategy of efficient heterogeneous catalysts for the reductive amination of FA lies in optimizing the geometric structure of the catalyst, the electronic density of the active metal species and the acid–base properties of the support. Moreover, the reductive amination of FF was predominantly performed in methanol rather than water as a solvent, and none of the studies covered in this review investigated the long-term stability of the catalyst. It is essential to understand the activity and deactivation of the catalysts under continuous flow conditions.

Note that except for using FF as the starting feedstock, Fernandes et al. [41] reported an interesting example of the direct one-pot synthesis of a large variety of FAM derivatives with good overall yields from xylose and xylan, catalyzed by an oxo-rhenium complex of $HReO_4$ using silanes as reducing agents. Up to now, however, there are still no successful reports on the heterogeneous catalytic conversion of xylose and xylan to FAM derivatives, owing probably to the complex transformation process.

**Table 1.** Reductive amination of FF with different nitrogen sources.

| Entry | Catalyst | Nitrogen Source | $PH_2$ (MPa) | Temp. (°C) | Time (h) | Conv. (%) | Yield (%) | Ref. |
|---|---|---|---|---|---|---|---|---|
| 1 | $Ru/Nb_2O_5$ | $NH_3$ gas | 4 | 90 | 4 | 100 | 99 | [43] |
| 2 | Ru-NPs | $NH_3$ in MeOH | 2 | 90 | 2 | 100 | 99 | [45] |
| 3 | Ru/BN-e | $NH_3$ aqueous | 1 | 90 | 5 | 100 | 99 | [46] |
| 4 | $4Ru_1Co/AC$ | $NH_3$ aqueous | 2 | 80 | 1 | 100 | 92 | [39] |
| 5 | $Ru_1/NC$-900-$800NH_3$ | $NH_3$ gas | 2 | 100 | 10 | 100 | 97 | [28] |
| 6 | Ru/TiP-100 | $NH_3$ gas | 1.7 | 30 | 24 | - | 91 | [33] |
| 7 | Raney Co | $NH_3$ gas | 1 | 120 | 2 | 100 | 99 | [51] |
| 8 | Co@C-600-EtOH | $NH_3$ in MeOH | 2 | 90 | 2 | 100 | 87 | [52] |
| 9 | Co/NC-700 | $NH_3$ in MeOH | 2 | 120 | 1 | 100 | 99 | [53] |
| 10 | $10Ni/Al_2O_3$ | $NH_3$ in MeOH | 2 | 100 | 2 | 100 | 92 | [54] |
| 11 | $Ni_1Al$ | $NH_3$ aqueous | 2 | 80 | 1 | 100 | 91 | [55] |
| 12 | $Ni/SiO_2$ | $NH_3$ gas | 2 | 90 | 1.5 | 100 | 98 | [56] |
| 13 | Ni/pNC | $NH_3$ gas | 4 | 60 | 6 | 100 | 99 | [57] |
| 14 | Ni@DS | $NH_3$ gas | 0.9 | 50 | 6 | - | 89 | [58] |
| 15 | $Ir/SiO_2$-$SO_3H$ | Aniline | 5 | 30 | 8 | 72 | 21 | [59] |
| 16 | $Rh_2P/NC$ | $HCOONH_4$ | 3 | 60 | 24 | - | 92 | [60] |

### 2.2. Reductive Amination of HMF

HMF, which is a fascinating molecule due to simultaneously containing three functional groups of aldehyde, alcohol and a furan ring, is produced by dehydration of hexoses (mainly fructose) or by hydrolysis/dehydration of cellulose in the presence of proper acidic catalysts [11]. It is known as a "sleeping giant" and is included together with FF as the top 10 value-added bio-based chemicals by the U.S. Department of Energy [62,63]. 5-(hydroxymethyl)-2-furfurylamine (HMFA) and its derivatives synthesized by the reductive amination of HMF are utilized in the manufacturing of bioactive compounds, including hypertension medications, diuretics and preservatives, and also as monomers in the synthesis of polymers, textiles and perfumes [64,65]. The reductive amination of HMF will be summarized and discussed on the basis of different nitrogen sources in this section.

#### 2.2.1. Reductive Amination of HMF with $NH_3$

The reductive amination of HMF with $NH_3$ is mainly based on non-noble Ni and Co catalysts. A Ni/SBA-15 catalyst was used by Chen et al. [66] for the reductive amination of HMF and ~90% yield of HMFA was obtained under the reaction conditions of 100 °C and 1.5 MPa $H_2$, which was significantly higher than that of Ru/C, Pd/C and Pt/C noble metal catalysts. They ascribed the high selectivity of the Ni/SBA-15 catalyst to its moderate hydrogenation activity. They also indicated that the presence of a small amount of water in the reaction system could preferentially promote the hydrolysis of HMFA to form ammonium ions, thus inhibiting the further condensation of HMFA with HMF to form secondary imines. In the catalyst life test, the Ni/SBA-15 catalyst showed slight deactivation after five consecutive runs, mainly caused by oxidation, aggregation, loss of nickel species and carbon deposition. Subsequently, a $Ni_6AlO_x$ nano-catalyst was prepared by Yuan et al. [67] showing an outstanding HMFA yield of 99% under reaction conditions of 100 °C and 0.1 MPa $H_2$. The partial encapsulation of the Ni and NiO nanoparticles by alumina in the $Ni_6AlO_x$ catalyst played an important role in the high selectivity of the catalyst. The catalyst could be recycled four times without any apparent loss of activity.

Room-temperature reductive amination of HMF over non-noble metal catalysts was first reported by Wang et al. [65]. They prepared an $Al_2O_3$-supported carbon-doped Ni catalyst by the direct pyrolysis-reduction of the mixture of $Ni_3(BTC)_2 \cdot 12H_2O$ and $Al_2O_3$

(Figure 4), and obtained a 96% yield of HMFA. By controlling the pyrolysis reduction temperature, the state of the Ni species and carbon-doping were finely adjusted to provide an air-stable metal $Ni^0$ species, which served as the active site in the reductive amination of HMF. Its remarkable performance was attributed to the synergistic effect of doped carbon, acidic $Al_2O_3$ support and air-stabilized metal $Ni^0$ species. Such a $Ni@C/Al_2O_3$-400 catalyst also demonstrated stable reusability and a wide range of substrate applicability.

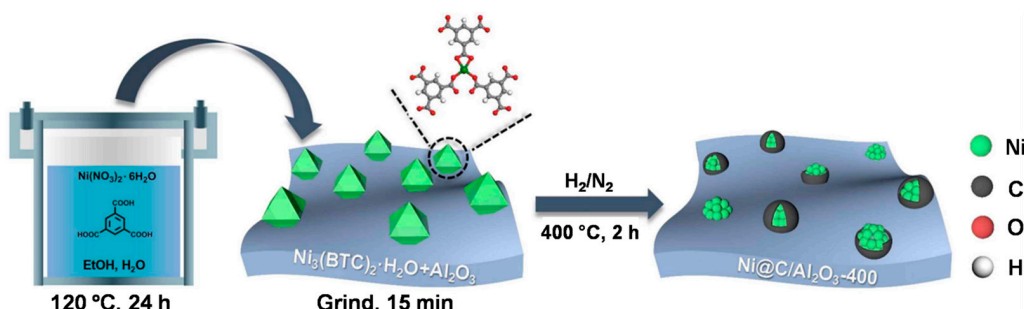

**Figure 4.** Schematic illustration for the synthesis procedure of a $Ni@C/Al_2O_3$-400 catalyst. Adapted with permission from ref. [65]. Copyright 2022 Wiley-VCH GmbH.

Besides nickel catalysts, cobalt-based catalysts were also used in the reductive amination of HMF with $NH_3$. Jagadeesh et al. [64] successfully prepared silica-supported Co nanoparticles by immobilizing and pyrolyzing a cobalt-terephthalic acid-piperazine MOF template on silica. Over this $Co-Co_3O_4@SiO_2$ catalyst, a 94% yield of HMFA was achieved under mild reaction conditions of 50 °C, 0.5 MPa $NH_3$ and 1 MPa $H_2$. However, the structure–activity relationship of the $Co-Co_3O_4@SiO_2$ catalyst has not been studied in depth.

### 2.2.2. Reductive Amination of HMF with Aniline

N-substituted HMFAs are a significant family of chemicals due to their well-known pharmacological properties. Iborra et al. [68] investigated the reductive amination of HMF with aniline over Pd-based catalysts with a similar particle size supported on activated carbon and alumina, respectively, and found that the Pd/C catalyst quantitatively converted HMF to the target N-phenyl-HMFA, while the $Pd/Al_2O_3$ showed low selectivity for N-phenyl-HMFA due to the hydrogenation of the furan ring. In the DRIFTS spectra of CO, only the unsaturated Pd atoms were observed over the Pd/C catalyst due to carbonaceous species deposited on the terraces. However, the existence of Pd (111) crystal planes in the $Pd/Al_2O_3$ catalyst provided a suitable interaction with the furan ring, leading to the hydrogenation of the furan ring. The Pd/C catalyst also presented high activity in the reductive amination of HMF with other amines and the one-pot reaction starting from nitrobenzene.

Queen et al. [69] developed a novel and highly stable MOF/polymer composite catalyst functionalized with Pd nanoparticles, defined as UiO-67/PpPDA/Pd, giving a high yield of 95% for N-phenyl-HMFA under mild reaction conditions of 50 °C and 0.5 MPa $H_2$. Even at room temperature, HMF could be quantitatively converted into the target amine with extended time. They ascribed the good activity and stability of the composite UiO-67/PpPDA/Pd catalyst to the synergy between the rigid structure of MOF and the high-density Lewis base of the polymer, which together inhibited the aggregation and leaching of Pd nanoparticles.

Cu-based catalysts have been used as active species for reductive amination because they favor the hydrogenation of C=N bonds over C=C bonds. A $CuAlO_x$ catalyst was used by Bukhtiyarov et al. [70] for the continuous synthesis of N-substituted HMFAs from the reductive amination of HMF in a flow reactor, and a 97% yield of N-phenyl-HMFA was obtained under reaction conditions of 100 °C and 1 MPa $H_2$, with methanol as a solvent to

facilitate the formation of imine intermediates. Over this low-cost Cu-based catalyst, other N-substituted HMFAs could also be produced with good yields.

Table 2 summarizes the recent studies on the reductive amination of HMF with different nitrogen sources. There are fewer reports on the reductive amination of HMF as compared with the reductive amination of FF and the catalysts are mainly focused on noble Pd and non-noble Ni, Co and Cu. Among the metal catalysts studied, Pd and Ni-based catalysts can be used for the reductive amination of HMF to synthesize HMFA and its derivatives under rather mild reaction conditions, even at room temperature. $NH_3$ gas and $NH_3$ in methanol are mainly used for the reductive amination of HMF, while $NH_3$ aqueous solution as a safe $NH_3$ resource is seldom used. Unlike the reductive amination of HMF with $NH_3$, non-noble Ni and Co-based catalysts were rarely applied in the reductive amination of HMF with aniline, while Cu-based catalysts were used in the latter reaction and presented a high yield toward the target amine product. The poor stability of the copper catalysts under the reaction conditions is still a challenge for sustainable process development. Developing bimetallic catalysts, such as NiCu and CoCu, may be a good choice for further improving the reductive amination activity and stability of copper-based catalysts.

**Table 2.** Reductive amination of HMF with different nitrogen sources.

| Entry | Catalyst | Nitrogen Source | $PH_2$ (MPa) | Temp. (°C) | Time (h) | Conv. (%) | Yield (%) | Ref. |
|-------|----------|-----------------|--------------|------------|----------|-----------|-----------|------|
| 1 | Ni/SBA-15 | $NH_3$ in MeOH | 1.5 | 100 | 4 | - | 90 | [66] |
| 2 | $Ni_6AlO_x$ | $NH_3$ gas | 0.1 | 100 | 6 | 100 | 99 | [67] |
| 3 | $Ni_1Al$ | $NH_3$ aqueous | 2 | 80 | 1 | 100 | 99 | [55] |
| 4 | Ni@C/$Al_2O_3$-400 | $NH_3$ in MeOH | 2 | 30 | 16 | 100 | 96 | [65] |
| 5 | Co-$Co_3O_4$@$SiO_2$ | $NH_3$ gas | 1 | 50 | 16 | 100 | 94 | [64] |
| 6 | Pd/C | Aniline | 0.3 | 100 | 1 | 100 | 100 | [68] |
| 7 | Pd/$Al_2O_3$ | Aniline | 0.3 | 100 | 1 | 100 | 95 | [68] |
| 8 | UiO-67/PpPDA/Pd | Aniline | 0.5 | 50 | 2 | - | 95 | [69] |
| 9 | $CuAlO_x$ | Aniline | 1 | 100 | 3 | - | 97 | [70] |

### 2.3. Reductive Amination of Other Bio-Based Furanic Aldehydes and Ketones

2.3.1. Reductive Amination of 2-Hydroxytetrahydropyran (2-HTHP)

Aminoalcohols containing both hydroxyl and amino functional groups are widely used in organic synthesis, especially as intermediates in the pharmaceutical and agro-chemical industries. For example, 5-Amino-1-pentanol (5-AP) is the starting material for the synthesis of alkaloid manzamine possessing anti-inflammatory and anti-cancer properties [71]. However, only very expensive test doses of 5-AP are currently available on the market. Recently, our group developed an alternative route for the more efficient and cleaner synthesis of 5-AP by the reductive amination of bio-FF-derived 2-HTHP (the tautomer of 5-hydroxyvaleraldehyde) over non-noble Ni-based catalysts (Route B, Figure 5) in comparison to the traditional route of chlorination and amination of 1,5-PD (Route A, Figure 5) [71,72].

**Figure 5.** The synthesis of 5-AP from different routes. Adapted with permission from ref. [73]. Copyright 2021 Royal Society of Chemistry.

Firstly, a variety of commonly used oxide supports including MgO, SiO$_2$, TiO$_2$, Al$_2$O$_3$ and ZrO$_2$ supported Ni catalysts prepared by the conventional impregnation (IM) method were investigated in the reductive amination of 2-HTHP [72]. The selectivity of the Ni catalysts toward 5-AP differed greatly (Table 3), which increased in the order of Ni/MgO < Ni/SiO$_2$ < Ni/TiO$_2$ < Ni/Al$_2$O$_3$ < Ni/ZrO$_2$. The highest 5-AP yield of ~85% was attained over the Ni/ZrO$_2$ catalyst at 80 °C and 2 MPa H$_2$, owing probably to the high reducibility and high surface acid density of the catalyst. However, an obvious deactivation was observed on the Ni/ZrO$_2$ catalyst after 22 h of operation in a fixed-bed reactor. The loss of active Ni due to leaching and surface oxidation caused this deactivation. In addition, hydroxyapatite (HAP) [74] and natural attapulgite (ATP) nanorods [75] were also explored as supports for the reductive amination of 2-HTHP to synthesize 5-AP. Higher yields of 92% and 94% to 5-AP were obtained over the 10Ni/HAP and 10Ni/ATP catalysts prepared by the precipitation–deposition (DP) method under similar conditions, respectively. The stability of these catalysts improved to some extent.

**Table 3.** Reductive amination of FF-derived 2-HTHP with NH$_3$.

| Entry | Catalyst | $PH_2$ (MPa) | Temp. (°C) | Time (h) | Conv. (%) | Yield (%) [a] | Lifetime (h) | Ref. |
|---|---|---|---|---|---|---|---|---|
| 1 | 15Ni/ZrO$_2$-IM | 2 | 80 | 1 | 100 | 85 (91) | <30 | [72] |
| 2 | 15Ni/Al$_2$O$_3$-IM | 2 | 80 | 1 | 100 | 81 | - | [72] |
| 3 | 15Ni/TiO$_2$-IM | 2 | 80 | 1 | 100 | 43 | - | [72] |
| 4 | 15Ni/SiO$_2$-IM | 2 | 80 | 1 | 99 | 21 | - | [72] |
| 5 | 15Ni/MgO-IM | 2 | 80 | 1 | 98 | 7 | - | [72] |
| 6 | 10Ni/HAP-DP | 2 | 80 | 1 | 100 | 92 | <60 | [74] |
| 7 | 10Ni/ATP-DP | 2 | 80 | 1 | 100 | 94 | 72 | [75] |
| 8 | 30Ni-MgAlO$_x$-CP | 2 | 60 | 1 | 100 | 83 (90) | <90 | [76] |
| 9 | 40Ni-MgAlO$_x$-CP | 2 | 60 | 2 | 100 | 93 | - | [76] |
| 10 | 50Ni-Al$_2$O$_3$-CP | 2 | 60 | 1 | 100 | 85 (91) | >150 | [73] |
| 11 | NiFe$_{0.25}$/Al$_2$O$_3$-CP | 2 | 80 | 1 | 100 | 90 | >120 | [77] |
| 12 | Ni$_5$Co$_1$-Al$_2$O$_3$-CP | 2 | 60 | 1 | 100 | 87 | >180 | [78] |

[a] The data shown in the parentheses were obtained under optimized conditions.

Subsequently, a more stable Ni-Mg$_3$AlO$_x$ catalyst with a hydrotalcite precursor structure was developed for the reductive amination of 2-HTHP to synthesize 5-AP [76]. The Ni-Mg$_3$AlO$_x$ catalyst with 30 wt% Ni loadings could maintain its 5-AP yield for 90 h continuous reaction, owing to the improved anti-sintering and anti-leaching stability by the confined effect of the hydrotalcite precursor structure. Nonetheless, the reaction activity decreased slowly with further reaction to 120 h, due mainly to the reorganization of the catalyst structure as the memory effect of the hydrotalcite catalyst precursor and the surface oxidation of active Ni$^0$ nanoparticles. To further improve the long-term stability of the catalyst, a Ni/Al$_2$O$_3$ catalyst with both a hydrotalcite and a spinel precursor structure was synthesized by using the coprecipitation (CP) method and optimizing the Ni loading or Ni/Al ratio [73]. The Ni/Al$_2$O$_3$ catalyst with around 50 wt% Ni loadings was found to present remarkable stability during a 150 h time-on-stream reaction without deactivation.

Compared with monometallic catalysts, bimetallic catalysts can significantly improve the selectivity and stability of catalysts due to many positive effects (size effect, electronic effect, structure effect, etc.) [47,48]. The incorporation of a second metal, such as Fe [77] or Co [78], into the Ni/Al$_2$O$_3$ was found to effectively promote the long-term stability of the monometallic Ni catalysts. The characterization results showed that the addition of an appropriate amount of the second metal resulted in the formation of a NiM (M = Fe or Co) alloy, which inhibited the sintering and surface oxidation of active Ni$^0$ particles during the continuous reductive amination of 2-HTHP.

The typical results for the synthesis of 5-AP from the reductive amination of FF-derived 2-HTHP are compiled in Table 3. High yields (>90%) of 5-AP were obtained over several none-noble Ni-based catalysts, owing to the matched hydrogenation activity and surface acid–basicity of the nano-Ni catalysts, which are superior to many commercial hydrogenation catalysts (including Raney Ni and the supported noble metals of Ru, Pd, Pt and Rh) [72,76]. Moreover, the strong metal-support interaction and the incorporation of appropriate promoters can improve the sintering resistance and surface oxidation resistance of the catalysts to enhance their long-term stability.

Note that more recently, Liu and coworkers reported on the synthesis of useful piperidine from bio-renewable tetrahydrofurfurylamine (produced from the reductive amination of FF or furfuryl alcohol with NH$_3$ and H$_2$ over Ni catalysts [79,80]) via its hydrogenolysis to the key reaction intermediate of 5-AP and the subsequent intramolecular amination of 5-AP over a Rh–ReO$_x$/SiO$_2$ catalyst [81]. The synergistic catalysis between the Rh nanoparticles and the ReO$_x$ species on the kinetically relevant cleavage of the C–O bond neighboring the C–NH$_2$ group in tetrahydrofurfurylamine contributed to the high efficiency of the catalyst. This work provides a new and efficient strategy for the synthesis of useful amines from biomass.

### 2.3.2. Reductive Amination of Cyclopentanone and 5-Diethylamino-2-Pentanone

Cyclopentanone is a pentacyclic ketone generated from the hydrogenation ring rearrangement reaction of furfural [6,82,83]. It can be further upgraded by reductive amination to a cyclopentamine product, an important fine chemical widely used as a starting synthetic material for pharmaceuticals, pesticides and cosmetics [14,84]. Wang et al. [84] used three different morphologies of niobium oxide-supported Ru-based catalysts for the reductive amination of cyclopentanone. The layered Nb$_2$O$_5$-L-supported Ru catalyst (Ru/Nb$_2$O$_5$-L) was found to exhibit the best catalytic performance, attaining an 84% yield of cyclopentamine under reaction conditions of 90 °C and 2 MPa H$_2$ (Table 4). They proposed that the Ru/Nb$_2$O$_5$-L catalyst possessed the highest surface area, resulting in the highest Ru dispersion, and therefore showed the highest catalytic activity. Over a partially reduced Ru/ZrO$_2$ catalyst, Zhang and coworkers [32] achieved up to a 93% yield of cyclopentamine in the reductive amination of cyclopentanone. Besides Ru-based noble catalysts, Ni/Al$_2$O$_3$ nano-catalysts were also found to exhibit high activity in the reductive amination of cyclopentanone to cyclopentamine. Up to a 97% yield of the target cyclopentamine could be

attained over the Ni/Al$_2$O$_3$ with around 50 wt% Ni loadings, which also presented good stability in reductive amination [73].

**Table 4.** The reductive amination of FF-derived cyclopentanone with NH$_3$.

| Entry | Catalyst | PH$_2$ (MPa) | Temp. (°C) | Time (h) | Conv. (%) | Yield (%) [a] | Ref. |
|-------|----------|--------------|------------|----------|-----------|---------------|------|
| 1 | Ru/Nb$_2$O$_5$-L | 2 | 90 | 4 | 100 | 84 | [84] |
| 2 | Ru/ZrO$_2$ | 1.2 | 90 | 12 | 100 | 93 | [32] |
| 3 | Ni/Al$_2$O$_3$ | 4 | 80 | 2 | 100 | 97 | [73] |

[a] The data shown in the parentheses were obtained under optimized conditions.

In addition to cyclopentanone, cyclopentenone derived from FF or HMF is also a versatile intermediate for the preparation of several naturally occurring products and bioactive molecules [85–88]. For example, diamino cyclopentenone is used for the synthesis of the cytotoxic marine sponge-derived alkaloid (±)-agelastatin A. Some heterogeneous catalysts, such as Montmorillonite K10 and Cu/SiO$_2$, have been reported sequentially for the direct conversion of FF to *trans*-4,5-diamino-cyclopentenone in almost quantitative yields [85,87].

Chloroquine phosphate, widely used to treat extraintestinal amoebiasis, malaria and rheumatism, has been a well-known antiviral drug on the market for many years [89,90]. $N^1,N^1$-diethyl-1,4-pentanediamine is the key side-chain of chloroquine phosphate, which is industrially produced via a two-step method of the condensation of 5-diethylamino-2-pentanone with ammonia and the subsequent hydrogenation with H$_2$ over a Raney Ni catalyst (Route A, Figure 6). This method has disadvantages, such as inherent low production efficiency of the batch reactor, potential safety hazards in the use of Raney Ni and liquid ammonia and the formation of large amounts of alkaline waste during the preparation of the Raney Ni catalyst. Recently, using active Ni$_x$Al nano-catalysts prepared by a simple coprecipitation method [55], our group highly selectively synthesized $N^1,N^1$-diethyl-1,4-pentanediamine through one-step reductive amination of 5-diethylamino-2-pentanone (Route B, Figure 6). Up to a 99% yield of the target product could be attained over the Ni$_1$Al catalyst, much higher than that of the commercial Raney Ni catalyst. The synergistic catalysis of highly dispersed active Ni$^0$ nanoparticles and abundant adjacent surface Lewis acidic sites accounted for the outstanding catalytic performance of the catalyst. Furthermore, this efficient catalyst exhibited remarkable stability during 200 h fixed-bed continuous operation.

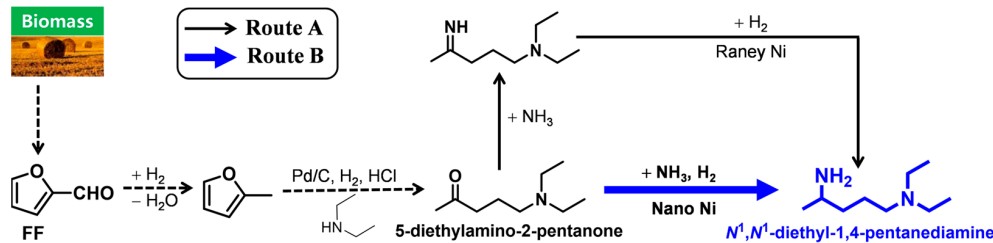

**Figure 6.** Reductive amination of FF-derived 5-diethylamino-2-pentanone and NH$_3$. Adapted with permission from ref. [55]. Copyright 2022 American Chemical Society.

### 2.3.3. Reductive Amination of 2,5-Diformylfuran (DFF)

2,5-bis(aminomethyl)furan (BAMF) has attracted much attention due to its valuable further valorization to products like polyamide and polyurethane for creating novel

biopolymers with distinctive properties [91,92]. BAMF can be obtained by reductive amination of DFF, a multifunctional platform chemical derived from the catalytic oxidation of HMF (Table 5).

**Table 5.** The reductive amination of DFF with $NH_3$.

| Entry | Catalyst | Nitrogen Source | $PH_2$ (MPa) | Temp. (°C) | Time (h) | Conv. (%) | Yield (%) | Ref. |
|-------|----------|-----------------|--------------|------------|----------|-----------|-----------|------|
| 1 | AT-Ni-Raney | $NH_3$ gas | 1 | 120 | 6 | 100 | 43 | [93] |
| 2 | Co/ZrO$_2$ | butylamine and $NH_3$ gas | 2 | 100 | 10 | 100 | 95 | [27] |

The synthesis of BAMF from DFF is challenging due to the easy polymerization of substrates and products. For example, Kim and coworkers [93] first used an acid-treated Raney Ni catalyst for the reductive amination of DFF with a 43% yield of BAMF under the optimum reaction conditions of 120 °C and 1 MPa $H_2$. They proposed a plausible reaction pathway (Scheme 3) for the reductive amination of DFF. Firstly, the two aldehyde groups of DFF reacted quickly with $NH_3$ to form the active primary imine intermediate, which was then hydrogenated to BAMF over the Raney Ni catalyst. Meanwhile, the reversible condensation of DFF and BAMF in the reaction system would form a ring or linear polymerized imine species. In the presence of excess $NH_3$, these imine species would be further converted to the target BAMF via ammonia-assisted hydrogenolysis. However, if these polymerized imines were hydrogenated, they would be converted into by-products of macrocyclic and linear amines, which were responsible for the low yield to BAMF. Therefore, the inhibition of the formation and hydrogenation of polymerized imine species plays a key role in the highly selective synthesis of BAMF from DFF.

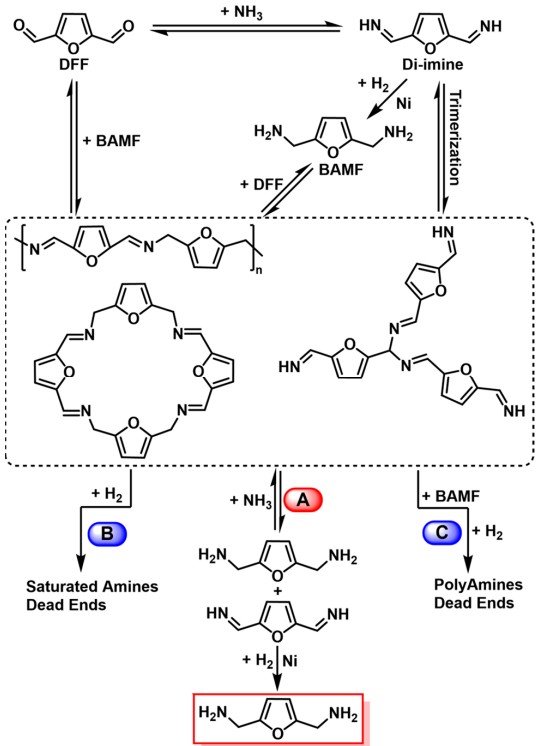

**Scheme 3.** Proposed reaction intermediates for the formation of BAMF with macrocyclic and linear polymers during the reductive amination of DFF. Adapted with permission from ref. [93]. Copyright 2015 Scientific Research Publishing.

The synthesis of BAMF from DFF by reductive amination required elaborate methods to prevent the formation of by-products, such as secondary amines, tertiary amines and polymeric amines. The group of Zhang and Wang [27] suggested that the concentrations of dialdehyde and primary diamine needed to be kept very low for inhibiting polymerization side-reactions. Given the higher nucleophilicity of alkylamines than ammonia and the reversibility of trans-imination, they proposed to use butylamine as an aldehyde scrubber and transition metal Co/ZrO$_2$ as an efficient catalyst to modulate the trans-imination and hydrogenation activity for highly selective synthesis of BAMF (Route B, Figure 7). Using this strategy, the polymerization side-reactions between dialdehyde and diamine were greatly inhibited and a high BAMF yield of 95% was achieved. This strategy can also be applied to the reductive amination of other biomass-derived dialdehydes, thus opening up a new path for bio-diamine monomers.

**Figure 7.** Reductive amination of DFF with NH$_3$ or butylamine. Adapted with permission from ref. [27]. Copyright 2020 Royal Society of Chemistry.

## 3. Reductive Amination of Levulinic Acid (LA)

LA is an important class of keto acid compounds derived from FF and HMF with both carboxyl and carbonyl groups [94,95] As a versatile chemical intermediate, it was included in the U.S. DOE's top 12 platform chemicals from biomass in 2004, and the revised and updated list in 2010. The cascade reductive amination and cyclization of LA is regarded as one of the most economical methods to prepare N-substituted 5-methyl-2-pyrrolidones (N-substituted 5-MePs) [15], which are used in various biological applications, including as precursors of piracetam and drug penetration enhancers [96]. For the reductive amination of LA to N-substituted 5-MePs, two reaction routes with imine (Route A) and amide (Route B) as the first intermediate, respectively, were proposed (Figure 8). In this section, the reductive amination of LA will be discussed on the basis of different nitrogen sources.

**Figure 8.** General reaction pathways for the reductive amination of LA with different primary amines.

### 3.1. Reductive Amination of LA with NH$_3$ or Primary Amines

Pioneering work for the reductive amination of LA to N-substituted 5-MePs under solvent-free conditions was performed by Shimizu and co-workers [97]. The authors reported that a Pt and MoO$_x$ co-loaded TiO$_2$ catalyst (Pt-MoO$_x$/TiO$_2$) was capable of smoothly catalyzing the reductive amination of LA with n-octylamine, affording up to a 99% yield of the target N-octyl-5-MeP under solvent-free and mild conditions of 100 °C and 0.3 MPa H$_2$. The high activity of the Pt-MoO$_x$/TiO$_2$ catalyst was attributed to the Lewis acidic sites of the catalyst, which promoted the reductive amination and subsequent

cyclization to generate the desired pyrrolidone. Subsequently, several noble catalysts were studied for the reductive amination of LA in the absence of a solvent. A Pt/TiO$_2$ catalyst was used by Sabater and coworkers [98] for the reductive amination of LA with aniline and nearly quantitative N-phenyl-5-MeP was obtained. They proposed that the Brönsted acidic sites generated by the spillover of dissociated hydrogen from the metal surface onto the TiO$_2$ support could promote the formation of imine intermediates. The reductive amination of LA with Pd/ZrO$_2$ as a catalyst under solvent-free conditions was also reported [99], and the yield of N-octyl-5-MeP exceeded 98% at 90 °C and 0.3 MPa H$_2$ with n-octylamine as the nitrogen source. The good catalytic performance of Pd/ZrO$_2$ catalysts was attributed to the strong Lewis acidity of the ZrO$_2$ supports, which could promote the reductive amination process and inhibit the side-reaction of the direct hydrogenation-esterification of LA.

The Pt-based catalysts above usually require high H$_2$ pressure (≥3 MPa) and temperature (≥90 °C). Therefore, it is highly desirable to develop efficient heterogeneous catalysts for the reductive amination of LA to N-substituted 5-MePs under ambient temperature and pressure. Han and co-workers [26] conducted pioneering work on the reductive amination of LA at ambient temperature and H$_2$ pressure using Pt nanoparticles supported on functional porous TiO$_2$ nanosheets (Pt/P-TiO$_2$) as an effective and stable heterogeneous catalyst. More than 30 examples showed the yields of N-substituted 5-MePs up to 90%. Among the various solvents investigated, methanol was the best solvent, as it facilitated the condensation of the carbonyl and amino groups and the subsequent cyclization process. They proposed that the porous nanosheet structure of Pt/P-TiO$_2$ was conducive to exposing more catalytic sites, the strong acidity of the P-TiO$_2$ support was favorable for the condensation of carbonyl and amino groups and the corresponding cyclization, and the low electron density of the metal Pt site was advantageous for the desorption of products and reactants. In another attempt by Liu et al. [100], various N-substituted 5-MePs were synthesized in 46–100% yields using cellulose-derived carbon (c-C) supported Pt nano-catalysts (Pt/c-C) under ambient conditions. The high performance of the catalyst was attributed to the cooperation between the c-C support and the Pt nanoparticles. The c-C support not only enhanced the electronic density of the Pt nanoparticles, but also provided acidic and basic sites for the Pt/c-C catalysts.

The reductive amination of LA in the presence of rare metal Ir-based catalysts has also been reported. The use of bifunctional metal–acid Ir/SiO$_2$-SO$_3$H as an efficient catalyst for the reductive amination of LA was studied by Martínez et al. [101], affording N-phenyl-5-MeP in a 63% yield. They proposed a plausible mechanism for the reductive amination of LA and aniline on an Ir/SiO$_2$-SO$_3$H catalyst that the interaction between the sulfonic group on the catalyst and the carboxylic group of LA (COOH-SO$_3$H) only made the carbonyl C2 available for nucleophilic addition to form the first intermediate of imine, which was then hydrogenated into the second intermediate of amine by H$_2$ adsorbed on the Ir nanoparticles. Finally, the second intermediate of amine spontaneously cyclized with the carboxyl group of LA to generate the desired N-phenyl-5-MeP (Scheme 4). Subsequently, Nagaoka and coworkers [102] performed the reductive amination of LA with aniline to N-phenyl-5-MeP on a polypyrrolidone stabilized Ir-PVP catalyst at room temperature, providing a 95% yield of N-phenyl-5-MeP. The Ir-PVP catalyst was stable for recycling three times without any loss of activity and was also applicable to the reductive amination of LA with nitroaromatic hydrocarbons/nitriles.

Despite great advances that have been made in noble metal catalysts, it is highly desirable to develop non-noble metal catalysts for the reductive amination of LA. Li et al. [103] successfully fabricated porous-carbon-coated Ni catalysts supported on carbon nanotubes (CNFx@Ni@CNTs) utilizing a facile atomic layer deposition (ALD) technique (Figure 9). The optimized CNF$_{30}$@Ni@CNTs catalyst achieved as high as a 99% yield of the target N-benzyl-5-MeP under the reaction conditions of 130 °C and 3 MPa H$_2$, and was recycled for 20 runs without apparent leaching or sintering of the Ni NPs under the shield of medium thickness porous carbon. Based on verification experiments and DFT calculations, the authors discovered that Ni-catalyzed reductive amination of LA followed an unusual

pathway with amide as the first intermediate, followed by tandem cyclization, intramolecular dehydration and hydrogenation to the desired N-benzyl-5-MeP. This mechanism was entirely distinct from the imine-intermediated route that was previously described in Pt-catalyzed systems. This research sheds light on the development of robust and functional non-noble metal heterogeneous catalysts, as well as the alteration of the reaction pathways to achieve the substitution of noble metals in the transformation of multifunctional bio-based substrates.

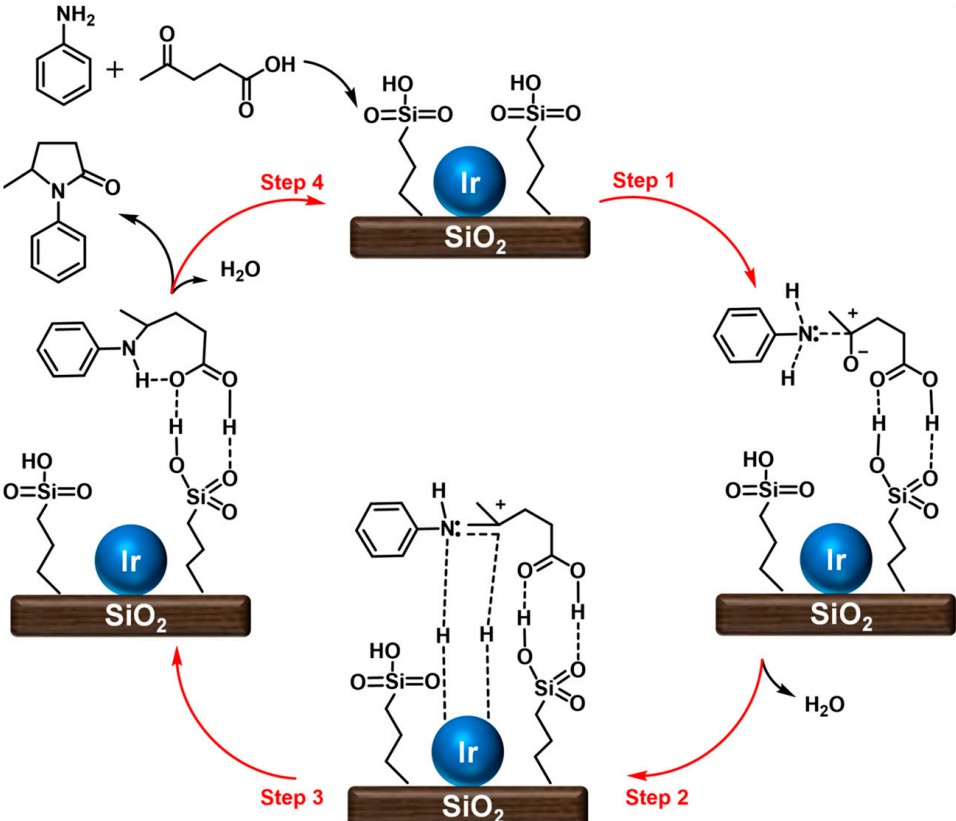

**Scheme 4.** Plausible mechanism for the reductive amination of LA with aniline over an $Ir/SiO_2$-$SO_3H$ catalyst. Adapted with permission from ref. [101]. Copyright 2017 Elsevier.

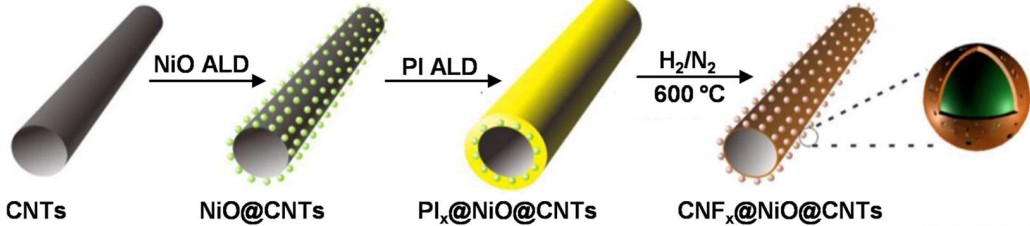

**Figure 9.** Schematic illustration for the synthesis procedure of a $CNF_x$@Ni@CNTs catalyst. Adapted with permission from ref. [103]. Copyright 2017 American Chemical Society.

Heterogeneous Cu-based catalysts have been attempted in reductive amination reactions; nonetheless, their poor thermal stability and short lifetime have limited their application to some extent. Some new strategies have been applied for creating stable Cu-based catalysts. For example, Chen and co-workers [104] constructed boron-doped $Cu/Al_2O_3$ catalysts ($Cu_{10}/AlB_3O$) for the reductive amination of LA with n-butylamine to N-butyl-5-MeP, achieving up to a 94% yield at 200 °C and 3 MPa $H_2$. Notably, the $Cu_{10}/AlB_3O$ catalyst also displayed good stability during the 200 h time-on-stream test

without discernible deactivation, owing to the strong interaction between the doped boron and copper species on the surface preventing the aggregation of Cu nanoparticles. A similar result was obtained on a copper-based bimetallic $Cu_{15}Pr_3/Al_2O_3$ catalyst [105]. The doping of an appropriate amount of lanthanide Pr into $Cu_{15}/Al_2O_3$ resulted not only in better dispersion and a smaller diameter of the copper species, but also in higher strong acidic sites, as well as larger specific surface areas of the catalyst, thus contributing to the high performance of the bimetallic catalysts, especially for $Cu_{15}Pr_3/Al_2O_3$.

Other non-noble bimetallic catalysts are also used for the reductive amination of LA. For instance, over a carbonized filter paper (CFP)-supported bimetallic FeNi alloy catalyst, Esposito et al. [106] achieved more than a 90% yield of N-phenethyl-5-MeP under the reaction conditions of 150 °C, 8.5 MPa $H_2$ and WHSV = 25 $h^{-1}$. The activity of the catalyst remained stable during the 52.5 h continuous operation test. By using chitosan as the nitrogen and carbon source for the synthesis of carbon-nitrogen-doped non-noble bimetallic catalysts (Co-M@Chitosan-X), Fang and co-workers [96] obtained a 93% yield of 5-MeP over a Co-Zr@Chitosan-20 catalyst under reaction conditions of 130 °C and 3 MPa $H_2$. Bimetallic synergy and carbon–nitrogen doping together contributed to the enhanced catalytic activity due to the strong electronic state and unique geometric structure.

### 3.2. Co-Reductive Amination of LA with Nitrile or Aldehyde

Given that nitriles or nitro compounds are precursors of primary amines, one-pot reductive amination of these compounds with LA provides a more cost-effective and step-economic method to synthesize N-substituted 5-MePs. Luque et al. [107] reported a one-pot reductive amination of LA with acetonitrile over the $Ru/TiO_2$ catalyst in a continuous flow reactor, attaining a 67% yield of N-ethyl-5-MeP under mild reaction conditions of 90 °C and 5 MPa $H_2$. Later, using a $MoO_x/TiO_2$-supported Pt catalyst, Shimizu et al. [108] performed the one-pot reductive amination of LA with n-octanonitrile to produce N-octyl-5-MeP with a yield of 92% under solvent-free and mild conditions of 110 °C and 0.7 MPa $H_2$. The Pt-$MoO_x/TiO_2$ catalyst also exhibited a broad range of substrate universality and good reusability.

The cascade and sequential one-pot synthesis of N-substituted 5-MePs from benzaldehyde and LA were carried out by Bhanage et al. [109] using a montmorillonite-supported nickel nano-catalyst (Ni-MMT). The sequential one-pot method was found to have a higher atom economy, resulting in a higher yield (86%) for the desired N-benzyl-5-MeP product than that obtained via the cascade route (63% yield). The sequential one-pot method is applicable for a wide range of double reductive amination processes of aldehyde and LA. Moreover, the Ni-MMT nanocomposite was stably recycled six times without deactivation.

To date, noble metals, such as Pt, Ir, Ru and Pd, and non-noble metals, such as Ni, Co and Cu, have been used for the reductive amination of LA to synthesize N-substituted 5-MePs, attaining high yields (up to 95% and above) of the target products (Table 6). A lot of work has been done to study the effects of metal compositions and particle sizes, bimetallic effects, catalyst surface properties, reaction solvents and reaction parameters on the reductive amination of LA. Due to the amine sources varying between different research works, it is a complex task to compare the activity of catalysts. In general, Pt-based catalysts present better catalytic activity in the reductive amination of LA, which can convert LA to N-substituted 5-MePs at ambient temperature and $H_2$ pressure. In the reaction path with imine as the first intermediate, the acidic sites of support and protic polar solvents can accelerate the condensation of carbonyl and amine functional groups and the subsequent cyclization. In addition, although the catalytic activity of LA performed better in a batch reactor, the reductive amination of LA in a continuous flow reactor has rarely been studied, which needs to be addressed in the future.

**Table 6.** Reductive amination of LA with different nitrogen sources.

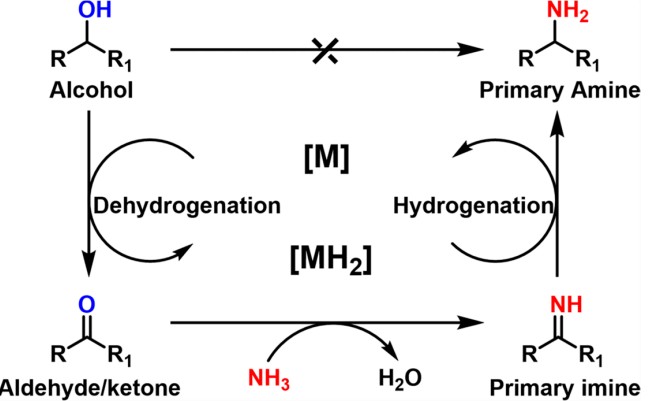

| Entry | Catalyst | Nitrogen Source | $PH_2$ (MPa) | Temp. (°C) | Time (h) | Conv. (%) | Yield (%) | Ref. |
|---|---|---|---|---|---|---|---|---|
| 1 | Pt-MoO$_x$/TiO$_2$ | n-Octylamine | 0.3 | 100 | 20 | - | 99 | [97] |
| 2 | Pt/TiO$_{2D}$ | Aniline | 0.5 | 150 | 18 | 100 | 100 | [98] |
| 3 | Pd/ZrO$_2$ | n-Octylamine | 0.5 | 90 | 12 | 99 | 98 | [99] |
| 4 | Pt/P-TiO$_2$ | n-Octylamine | 0.1 | 30 | 3 | - | 97 | [26] |
| 5 | Pt/c-C | Aniline | 0.1 | 30 | 3 | - | 96 | [100] |
| 6 | Ir/SiO$_2$-SO$_3$H | Aniline | 3.5 | 100 | 8 | 63 | 63 | [101] |
| 7 | Ir-PVP | Aniline | 0.5 | 30 | 24 | 99 | 95 | [102] |
| 8 | CNF$_{30}$@Ni@CNTs | Benzylamine | 3 | 130 | 6 | - | 99 | [103] |
| 9 | Cu$_{10}$/AlB$_3$O | n-Butylamine | 3 | 200 | - | 99 | 94 | [104] |
| 10 | Cu$_{15}$Pr$_3$/Al$_2$O$_3$ | n-Butylamine | 5 | 175 | 20 | 100 | 94 | [105] |
| 11 | FeNi/C | Phenethylamine | 8.5 | 150 | - | 93 | 93 | [106] |
| 12 | Co-Zr@Chitosan-20 | NH$_3$ aqueous solution | 3 | 130 | 24 | 94 | 93 | [96] |
| 13 | Ru/TiO$_2$ | Acetonitrile | 5 | 90 | - | 79 | 67 | [107] |
| 14 | Pt-MoO$_x$/TiO$_2$ | n-Octanenitrile | 0.7 | 110 | 24 | 100 | 92 | [108] |
| 15 | Ni-MMT | Benzaldehyde | 1.5 | 140 | 16 | - | 86 | [109] |

## 4. Hydrogen-Borrowing Amination of Bio-Based Furanic Alcohols

The plentiful biomass-based alcohols derived from non-edible lignocellulose are ideal candidates for the sustainable production of amine chemicals. The amination of alcohols to amine can be realized by the so-called hydrogen-borrowing mechanism, also known as the automated hydrogen transfer method, which includes three consecutive steps of dehydrogenation, amination and hydrogenation (Scheme 5) [110]. Specifically, these steps are (i) the dehydrogenation of alcohols to highly reactive aldehydes (or ketones), (ii) the condensation of carbonyl compounds with ammonia to form imines and (iii) the hydrogenation of the imines to form amines over the active sites of the catalyst. Theoretically, no additional H$_2$ is consumed and water is the only by-product, which highlights the advantages of this route with high atomic economy and environmental protection [79,111]. However, high reaction temperatures (150~250 °C) are usually required in the amination of alcohols to activate the C–O bond of alcohols, which is thought of as a rate-determining step. High temperatures will lead to an increase in side-reactions, such as cyclization, overalkylation and polymerization [112]. Therefore, it is important to find an appropriate balance between activity (i.e., the conversion of C–OH bonds) and selectivity (i.e., the avoidance of unwanted side-reactions) by designing specific heterogeneous catalysts.

**Scheme 5.** Amination of alcohols with NH$_3$ by the hydrogen-borrowing mechanism.

### 4.1. Hydrogen-Borrowing Amination of Furfuryl Alcohol (FA)

FA is one of the most prevalent products of FF hydrogenation and ~62% of the FF produced globally each year is estimated to be converted into FA [113]. In addition to the reductive amination of FF, the amination of FA also represents one of the potential pathways for the synthesis of FAM through a hydrogen-borrowing amination mechanism. Some heterogeneous catalysts containing Ni, Co or Ru are currently available for the amination of FA to FAM.

Hara et al. [114] used a Ru-20MgO/TiO$_2$ catalyst for the amination of FA with a 94% yield of FAM in the absence of H$_2$. They proposed a reaction mechanism that the alcohol was adsorbed on the Ru surface to form deprotonated Ru alkoxide, followed by β-hydride elimination to generate aldehydes. After that, the reversible condensation of the resulting aldehydes with NH$_3$ formed the imines, which were hydrogenated on the concomitant Ru-H species to produce the desired primary amines (Scheme 6). The addition of MgO to the catalyst not only provided electrons for metal Ru, but also promoted the high dispersion of Ru nanoparticles.

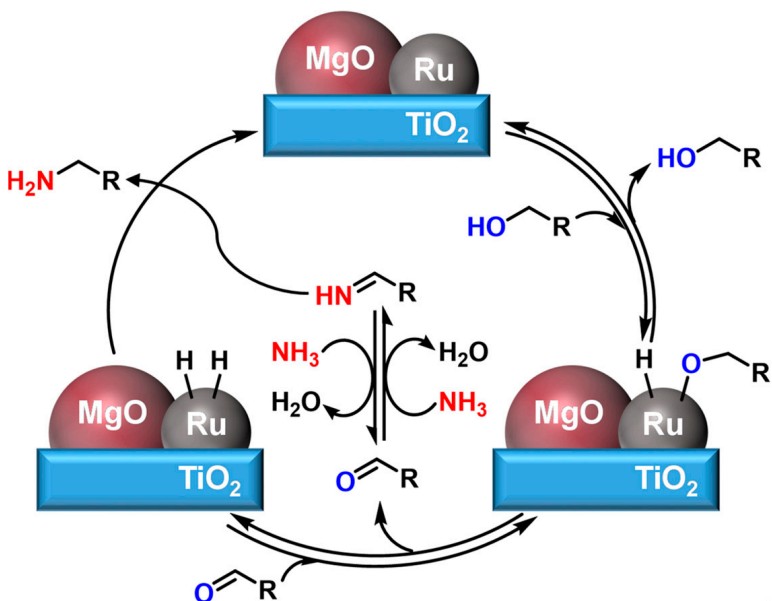

**Scheme 6.** Proposed reaction mechanism for the amination of alcohols over Ru-20MgO/TiO$_2$. Adapted with permission from ref. [114]. Copyright 2020 Royal Society of Chemistry.

Besides noble metal Ru, non-noble metal Ni-based catalysts are also active in the amination of FA. Wei et al. [79] used commercial Raney Ni for the amination of FA, obtaining a 77% yield of FAM in the absence of H$_2$. They ascribed the formation of FAM to the proper adsorption capacity of Raney Ni for NH$_3$, H$_2$ and FA molecules. However, the formation of a new Ni$_3$N phase led to the deactivation of the catalyst. It should be noted that Raney Ni could be recycled at least five times without noticeable deactivation in the presence of 1 MPa H$_2$.

Subsequently, using the Ni$_x$Al catalyst with different Ni/Al mole ratios prepared from a hydrotalcite precursor, Xu et al. [115] carried out the amination of both FA and HMFA to the corresponding FAM and BAMF products. Over a Ni$_2$Al-600 catalyst reduced at 600 °C, an 84% yield of FAM and a 71% yield of BAMF were achieved, respectively. The cooperative effect of its abundant Ni$^0$ sites and appropriate acid–base site density is suggested as an important parameter for the catalyst with good catalytic activity (Scheme 7). However, the formation of Ni$_3$N during the recycling tests led to the gradual deactivation of the catalyst.

**Scheme 7.** A possible mechanism of synergistic catalysis of $Ni^0$ and acid-base sites for the amination of FA. Adapted with permission from ref. [115]. Copyright 2021 Chemistry Europe.

The typical results of the synthesis of FAM from the amination of FA are presented in Table 7. It was obvious that the amination of FA to FAM was carried out through the mechanism of hydrogen-borrowing amination in absence of $H_2$, and the heterogeneous catalysts employed were focused on the noble metal Ru and non-noble metal Ni. In light of the FAM yield and reaction temperature, the Ru-based catalyst is more active than the Ni-based catalyst. Moreover, the Ni-based catalyst displayed poor stability in the continuous cycle test due to the formation of a new $Ni_3N$ phase. Therefore, it is necessary to develop novel Ni-based catalysts to achieve high yield and sustainable long-term synthesis of FAM.

**Table 7.** Hydrogen-borrowing amination of FA with $NH_3$.

| Entry | Catalyst | Nitrogen Source | $PH_2$ (MPa) | Temp. (°C) | Time (h) | Conv. (%) | Yield (%) | Ref. |
|-------|----------|-----------------|--------------|------------|----------|-----------|-----------|------|
| 1 | Ru-20MgO/TiO$_2$ | NH$_3$ gas | 0 | 110 | 20 | 100 | 94 | [114] |
| 2 | Raney Ni | NH$_3$ gas | 0 | 180 | 24 | 79 | 77 | [79] |
| 3 | Ni$_2$Al-600 | NH$_3$ gas | 0 | 180 | 36 | - | 84 | [115] |

*4.2. Hydrogen-Borrowing Amination of Bio-Based Furanic Alkanediols*

Alkanediols with five and six carbon atoms generated from FF and HMF are versatile intermediate platform molecules [116,117]. The amination of diols represents an efficient method for biomass valorization to value-added chemicals due to the importance of amines, including aminoalcohols, diamines and N-heterocyclic amines, in the polymer industry [18,20]. More efforts have been made to convert biomass-derived diols into amines, particularly straight-chain diamines, in light of the rising demand for biopolymers.

4.2.1. Hydrogen-Borrowing Amination of 1,5-Pentanediol (PDO)

Research on the amination of PDO over heterogeneous catalysts is quite limited. Kulkarni et al. [118] used a Co-modified ZSM-5 catalyst for the amination of PDO with 87%

conversion and a 43% yield toward 5-AP (Figure 10). Moreover, its stability under reaction conditions needs to improve, and the structure–activity relationship of the catalyst needs to be further elucidated.

**Figure 10.** Hydrogen-borrowing amination of PDO and $NH_3$ to synthesize 5-AP.

The heterogeneous catalysts currently used for the di-amination of PDO give a cyclic piperidine product instead of 1,5-pentanediamine because the direct cyclization of the intermediate 5-AP is thermodynamically more favorable than further hydrogen-borrowing amination (Figure 11). For example, Rose et al. [119] studied the amination of PDO with $NH_3$ into piperidine (86% yield) as the main product over a solid Ru/C catalyst in an aqueous solution.

**Figure 11.** Hydrogen-borrowing amination of PDO and $NH_3$ to synthesize piperidine.

### 4.2.2. Hydrogen-Borrowing Amination of 1,6-Hexanediol (HDO)

1,6-Hexanediamine (HDA), a key monomer in the synthesis of nylon-66, has traditionally been obtained by the hydrogenation of adiponitrile produced with butadiene and toxic HCN [120]. Many efforts have been made to achieve a green and sustainable synthesis of HDA by the amination of renewable HDO derived from biomass.

Zhao and coworkers [121] performed the amination of HDO over a $Ru/Al_2O_3$ catalyst with a 38% yield of HDA in the presence of supercritical $NH_3$. The high dispersion of Ru and medium acid–basicity in the catalyst are crucial factors for its good activity and selectivity. The $NH_3$ with 15 MPa at a supercritical state inhibited side-reactions, such as cyclization, dimerization and oligomerization, thus improving the selectivity of the target product.

The amination of HDO is usually carried out at high temperatures and $NH_3$ pressures, which significantly hinder the actual large-scale amination of HDO. Recently, Rose et al. [122] used a Ru/C catalyst for the amination of HDO to aminoalcohols or diamines under the reaction conditions of 190 °C, 2.5 MPa $H_2$ and $NH_3$ aqueous solution. In the presence of $Cs_2CO_3$, a 26% yield of 6-amino-1-hexanol was achieved. On the other hand, a 34% yield of HDA was obtained in the presence of $Ba(OH)_2$. They concluded that the presence of the base facilitated the dehydrogenation of alcohols and the condensation of carbonyl intermediates with $NH_3$.

Tetramethyl diamines, an industrially important class of fine chemicals, are often produced from alkyl halides with large salt-containing waste. The amination of diols with dimethylamine represents one of the potential pathways for producing tetramethyl diamines. Yan et al. [123] developed a $Cu/ZnO/\gamma\text{-}Al_2O_3$ catalyst for the amination of HDO with dimethylamine to produce N,N,N′,N′-tetramethyl-1,6-hexanediamine (TMHDA) in a fixed-bed reactor. Over this $Cu/ZnO/\gamma\text{-}Al_2O_3$ catalyst, a 93% yield of TMHDA was achieved under reaction conditions of 180 °C and 3 MPa $H_2$. The doped ZnO could effectively improve the dispersion of active Cu and reduce the particle size of Cu, owing to the strong interaction between Cu and ZnO, which help the catalyst expose more active sites and thus promote the catalytic activity.

N-alkyl amines are extensively used in the production of various materials, pharmaceuticals and pesticides. Shi and coworkers [124] used a non-noble $CuNiAlO_x$ catalyst

synthesized by the coprecipitation method for the amination of C4-C6 linear diols to produce aminoalcohol or diamine products. At low catalyst concentrations, aminoalcohol was obtained as the main product. At high catalyst concentrations, the conversion of diol to dialdehyde was enhanced, followed by dehydration to form an imine and hydrogenation to form a cyclization by-product. Additionally, the presence of abundant amine could quickly react with the in-situ generated dialdehyde, which prohibited the cyclization side-reaction and promoted diamine production.

Table 8 presents the results of studies on the amination of HDO to produce HDA and its derivatives over various heterogeneous catalysts. As described above, for the synthesis of HDA, the yield is less than 50% due to the easy conversion of HDO to a stable seven-membered N-heterocyclic by-product, whereas the synthesis of tertiary amines derived from HDO is relatively easy to obtain in high yields, as there is no cyclization side-reaction during the reaction. In addition, although hydrogen is not consumed, certain hydrogen pressure is still required to maintain the catalytic activity in the amination of diols described above. Therefore, more efforts are required for the synthesis of straight-chain primary diamines from bio-based furanic diols.

**Table 8.** Hydrogen-borrowing amination of HDO with different nitrogen sources.

| Entry | Catalyst | Nitrogen Source | $PH_2$ (MPa) | Temp. (°C) | Time (h) | Conv. (%) | Yield (%) | Ref. |
|-------|----------|-----------------|--------------|------------|----------|-----------|-----------|------|
| 1 | Ru/Al$_2$O$_3$ | Supercritical NH$_3$ | 2 | 220 | 6 | 100 | 38 | [121] |
| 2 | Ru/C | NH$_3$ aqueous solution | 2.5 | 190 | 2 | - | 34 | [122] |
| 3 | Cu/ZnO/γ-Al$_2$O$_3$ | Dimethylamine | 3 | 180 | - | 100 | 93 | [123] |
| 4 | CuNiAlO$_x$ | Aniline | 0 | 180 | 24 | 100 | 89 | [124] |

*4.3. Reductive Amination and Hydrogen-Borrowing Amination of HMF to Synthesize BAMF*

As mentioned in the section on the reductive amination of DFF, BAMF is an emerging monomer for various bio-based polymers, such as polyurethane, polyamine and polyurea [91,92,125]. Except for the reduced amination of DFF, the direct amination of HMF provided an alternative pathway for the synthesis of BAMF. As shown in Figure 12, many multi-step routes for the synthesis of BAMF from HMF have been developed [27,43,93,126,127]. The direct amination of HMF to BAMF is possibly preferable to multi-step procedures.

**Figure 12.** Catalytic synthesis of BAMF from HMF. Adapted with permission from ref. [128]. Copyright 2011 RSC Publishing.

Based on the high dehydrogenation performance of copper and the good hydrogenation performance of nickel, Shi et al. [128] developed a simple and efficient bifunctional

non-noble $CuNiAlO_x$ catalyst for the conversion of HMF to BAMF in a one-pot process. Owing to bulk Cu and highly dispersed Ni species, BAMF with a yield up to 86% was attained over $Cu_4Ni_1Al_4O_x$ in the presence of a $Na_2CO_3$ co-catalyst for promoting the dehydrogenation reaction.

Wei et al. [80] investigated different commercial catalysts for the conversion of HMF to BAMF and obtained a 61% BAMF yield over Raney Ni under the conditions of 160 °C, 0.35 MPa $NH_3$ and 1.0 MPa $H_2$. The DFT calculation reveals that the difference in the adsorption energies of metal Ni to $NH_3$ and $H_2$ is much lower than that of other metals, which reduces ammonia coverage on the catalyst surface, resulting in more vacancy active centers to adsorb and activate hydrogen. Later, they [129] achieved up to an 86% yield of BAMF over $\gamma$-$Al_2O_3$-supported Ni catalysts with around 10 wt% Ni loadings under the same conditions as Raney Ni. The incorporation of a proper amount of Mn enhanced the reaction stability of the catalyst.

The direct conversion of HMF to BAMF has only been accomplished by some Ni-based heterogeneous catalysts (Table 9). It is known that Ni-based catalysts are susceptible to deactivation in the amination of carbonyl or alcohol-based oxygenates to amine products due to the accumulation of metal nitrides, alkylamines and carbon-containing compounds on the catalyst surface, as well as the oxidation and leaching of active metal species. However, the stability of these Ni-based catalysts under continuous flow conditions has not been investigated in this section. Therefore, it is necessary to understand the deactivation of the catalyst under flow conditions for examining the feasibility of their large-scale research and commercialization and for developing new heterogeneous catalysts for achieving a high yield and a sustainable long-term synthesis of BAMF at low temperatures and pressures.

**Table 9.** Direct amination of HMF with $NH_3$ to synthesize BAMF.

| Entry | Catalyst | Nitrogen Source | $PH_2$ (MPa) | Temp. (°C) | Time (h) | Conv. (%) | Yield (%) | Ref. |
|---|---|---|---|---|---|---|---|---|
| 1 | $Cu_4Ni_1Al_4O_x$ | $NH_3$ gas | 4.5 | 210 | 18 | 100 | 86 | [128] |
| 2 | Raney Ni | $NH_3$ gas | 1 | 160 | 12 | 100 | 61 | [80] |
| 3 | $10Ni/\gamma$-$Al_2O_3$ | $NH_3$ gas | 1 | 180 | 36 | 100 | 86 | [129] |
| 4 | $10NiMn(4:1)/\gamma$-$Al_2O_3$ | $NH_3$ gas | 1 | 160 | 24 | 100 | 82 | [129] |

## 5. Conclusions and Prospects

The synthesis of useful amines from abundant and renewable bio-based furanic oxygenates via reductive amination or hydrogen-borrowing amination has attracted much attention in recent years. In this review, the latest advances in the production of high-value-added amines from the amination of bio-based furanic oxygenates with $H_2$ and different nitrogen sources over heterogeneous catalysts are critically highlighted. To attain these valuable amines with high yields, the fabrication and catalytic performances of heterogeneous noble metal (Ru, Pd, Pt, Rh) and non-noble metal (Ni, Co, Cu) catalysts with different specific structures are comprehensively evaluated from three aspects of catalytic activity, selectivity and stability. The key factors affecting the performance of these catalysts, such as active metals, supports, promoters, reaction solvents and conditions, are studied and discussed in-depth. It is crucial to understand the structure–activity relationships and reaction mechanisms for the development of novel heterogeneous catalysts, which are also deeply analyzed by combining advanced characterization and theoretical calculation. Despite numerous efforts that have been undertaken, the following key points still need to

be considered to make the synthesis of sustainable amine chemicals from biomass-based oxygenates more refined and valuable.

### 5.1. Development of the Catalyst

For the reductive amination of bio-based furanic aldehydes and ketones, the reaction conditions are relatively mild, and even the corresponding amines with a nearly quantitative yield can be attained at ambient temperature and $H_2$ pressure over noble Ru, Pd and Pt, and non-noble Ni-based catalysts. Nonetheless, for hydrogen-borrowing amination of bio-based furanic alcohols, harsh reaction conditions, i.e., reaction temperatures as high as 160–250 °C, are usually required because the kinetically relevant dehydrogenation of alcohol hydroxyl to highly active carbonyl is an endothermic reaction with high activation energy. Many side-reactions, such as overalkylation, cyclization and polymerization, occur under such reaction conditions, making it difficult to achieve a high yield of the target amine. At present, good to excellent yields of the target monoamines are facilely achieved, whereas the yields of diamines, particularly straight chain primary diamines, are rather low (usually below 50%), owing to the generation of many by-products. Therefore, it is highly desirable to adopt some effective strategies, such as forming a highly active alloy or polymetallic active sites, regulating the surface acid–basicity of the catalyst by using suitable supports and additives, and building a confined environment with pore, interlayer or shell structures to prepare novel catalysts for achieving high yield and good stability in synthesizing the target amines, especially primary diamines.

In addition, despite the fact that noble metals like Ru and Pt-based catalysts show high activity in the amination of the bio-based furanic oxygenates, the high price and scarcity of these metals restrict their large-scale application. Non-noble metals, such as Ni and Co-based catalysts, are more attractive, owing to their high potential for industrial applications in the amination of bio-based furanic oxygenates. However, the stability of these catalysts is usually under question due to the accumulation of metal nitrides, alkylamines and carbon-containing compounds on the catalyst surface, as well as the oxidation and leaching of the active metal species. Improvements regarding the stability, particularly at high metal loading and large-scale catalyst synthesis, need to be addressed before a practical use of these non-noble metal catalysts can be employed.

### 5.2. Optimization of Reaction Conditions and Process

To date, most of the amines synthesized from bio-based furanic oxygenates have been carried out in the presence of a solvent, with the exception of the reductive amination of LA. Organic solvents like methanol are frequently used, which not only increase the cost, but also raise the issue of safety and environmental pollution. Therefore, it is necessary to develop heterogeneous catalysts with high activity and stability in green solvents like water or even solvent-free environments.

Moreover, most of the reported reductive amination and hydrogen-borrowing amination of bio-based furanic oxygenates are performed in batch reactors on a milligram or gram scale, and further utilization of these data in the industry will be under question. The use of continuous reactors at a larger scale to investigate the activity and long-term stability of catalysts would undoubtedly be beneficial for the industrial production of the target amines.

In addition, the "one-pot" method of multi-stage temperature subsection control needs to be considered for upgrading bio-based furanic oxygenates to target amines, which will minimize the manufacturing costs, improve the yield of the required amines, and simplify the separation of the required products. The integration of the in-situ generation of the highly reactive bio-based furanic oxygenates and their further amination provides a very attractive but challenging method for the effective synthesis of the target amines. Special attention needs to be paid to the rational design of the multi-functional catalyst system and reaction process.

*5.3. Studies on Reaction Mechanism and Kinetics*

For various catalytic systems in the amination of bio-based furanic oxygenates, many different types of reaction pathways have been proposed. However, an in-depth understanding of the catalytic reaction mechanism is still lacking, which limits the rational design of catalysts and the modulation of kinetic reaction parameters for amination reactions. To gain insight into the structure–activity relationships and reaction mechanisms, it is necessary to combine in situ spectroscopic techniques such as EXAFS, in situ IR and solid-state NMR, as well as reaction kinetic studies with relevant theoretical calculations. This will contribute to an in-depth understanding of the adsorption and activation patterns of different substrates (including oxygenates, amines and hydrogen), the formation and further conversion of the reaction intermediates, and the generation and desorption of both the target and by-products during dynamic and realistic heterogeneous catalytic processes. These studies will help to provide the experimental and theoretical basis for the development of novel stable catalysts with high efficiency in the amination of special oxygenates with multifunctional groups at the molecular or atomic level.

**Author Contributions:** Conceptualization, J.Z. and Z.H.; methodology, J.Z. and J.Y.; writing—original draft preparation, J.Z., J.Y., X.L. and X.Y.; writing—review and editing, J.Z., H.L. and Z.H.; funding acquisition, H.L., C.X. and Z.H. All authors have read and agreed to the published version of the manuscript.

**Funding:** This work was funded by the National Natural Science Foundation of China (21872155, 22102198, 22272187), the Strategic Pilot Science and Technology Project of the Chinese Academy of Sciences (XDA21010700) and the CAS "Light of West China" Program.

**Data Availability Statement:** No new data were created or analyzed in this study. Data sharing is not applicable to this article.

**Conflicts of Interest:** The authors declare no conflict of interest.

**Abbreviations**

| | |
|---|---|
| FF | furfural |
| HMF | 5-hydroxymethylfurfural |
| FAM | furfurylamine |
| SACs | single-atom catalysts |
| NPs | nanoparticles |
| HMFA | 5-(hydroxymethyl)-2-furfurylamine |
| 2-HTHP | 2-hydroxytetrahydropyran |
| 5-AP | 5-amino-1-pentano |
| HAP | hydroxyapatite |
| ATP | attapulgite |
| IM | impregnation |
| DP | deposition–precipitation |
| CP | coprecipitation |
| DFF | 2,5-diformylfuran |
| BAMF | 2,5-bis(aminomethyl)furan |
| LA | levulinic acid |
| 5-MeP | 5-methyl-2-pyrrolidone |
| ALD | atomic layer deposition |
| FA | furfuryl alcohol |
| PDO | 1,5-pentanediol |
| HDO | 1,6-hexanediol |
| HAD | 1,6-Hexanediamine |
| TMHDA | N,N,N′,N′-tetramethyl-1,6-hexanediamine |

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
