# Peer review of "Recent Advances in the Efficient Synthesis of Useful Amines from Biomass-Based Furan Compounds and Their Derivatives over Heterogeneous Catalysts"

_catalysts, doi:10.3390/catal13030528_

Round 1
Reviewer 1 Report
Liu, Huang and co‑workers present a review paper concerning the conversion of biomass‑derived compounds, namely furfural, hydroxymethylfurfural and derivatives thereof. In particular, the manuscript is focused on the catalytic systems (actually, it seems to be particularly referred to transition‑metal based catalytic systems) devoted to the reductive amination of such compounds to obtain amines, whose importance as building blocks for organic synthesis is generally recognized. Despite the presence in the literature of review papers facing the same topic, like Green Chem. 2020, 22, 6714 (which is also ref.11 of the present manuscript under review), the paper appears to be up‑to‑date, and several works from 2020, 2021 and 2022, not covered by previous review papers, have been correctly taken into account (even if many of these works have been carried out by the some of the authors). However, in many sentences the english is really bad, lots of typos are still present and a careful check by an english native speaker is mandatory. Furthermore, some conceptual mistakes and other points to clarify are still present. The suggestion of this reviewer is to postpone the evaluation of the present review paper after its substantial revision. More attention to typos from the authors is highly recommended before submitting the revised manuscript.
GRAMMAR AND STYLE
1) Page 1, line 10: “nitrogen source” should be turned to its plural form “nitrogen sources”
2) Page 1, line 11: “building block” should be turned to its plural form “building blocks”
3) Page 1, line 12: “challenging” what? The object for this adjective is missing!
4) Page 1, line 13: “side reaction” should be turned to its plural form “side reactions”
5) Page 1 line 18: as for page 1, line 10
6) Page 2, line 49: the sentence “to green synthesize of high value‑added chemicals” should be better rendered in english. A possible suggestion from this reviewer is “to synthesize high value‑added chemicals in a green fashion”
7) Page 2, line 54: “which employed” must be corrected by eliminating “which”
8) Page 2, line 60: “In general” is superfluos, and the authors may efficiently start the sentence with “Traditionally”
9) Page 2, line 63: Once again, “theoretically” is superfluos, since later on in the sentence the authors use “in theory”
10) Page 3, line 84: “generation OF water”
11) Page 3, line 90: the english grammar of the last part of the sentence is really bad. This reviewer suggests to correct in something like: “likely being the rate‑determining step”
12) Page 3, line 98: Once again, really bad english: “homolytic or heterolytic activation” is the correct form if “activation” is the subject. Another possible correct form is “homolysis or heterolysis of the H‑H bond in molecular H2”: in this case, homolysis or heterolysis” are the subjects of the sentence, rather than two adjectives
13) Page 4, line 114: “acidic catalysts”, where “acidic” is an adjective
14) Page 4, line 118: “FF can be facilely reductive amination” has totally no sense in english grammar! Please replace with “FF can be facilely undergo reductive amination to afford….”
15) Page 4, line 125: the verbal time needs to be corrected, the section is still going to be discussed from the perspective of the reader (“the reductive amination of FF will be discussed/is discussed”)
16) Page 5, line 155: “Ru‑based catalytic NPs” is more correct from the perspective of grammar (the same observation is valid later on in the manuscript)
17) Page 6, line 182: “Differently” rather than “different”
18) Page 6, line 200: Challenging is an adjective! Really bad english grammar! Please reformulate!
19) Page 7, lines 252‑253: “in the synthesis of FAM” or “in synthesizing FAM” are the correct forms
20) Page 7, line 271: water is “(polar and protic)”
21) Page 8, line 300: “Rh2P/NC catalyst could also be used”: this sentence is related to a literature reference, its generality has not been proved and, therefore, the verb must be referred to the past
22) Page 9, line 309: are the authors sure that “wildly” is an appropriate adverb in this context?
23) Page 9, line 314: “investigated the long‑term stability of the catalyst”
24) Page 9, line 317‑320: The structure of the sentence is completely wrong from a formal point of view, and contains furthermore some typos. It must be corrected in the form: “HMF, which is a fascinating molecule due to simultaneously containing three functional groups of aldehyde, alcohol and furan ring, is produced by dehydration of hexoses (mainly fructose) or by hydrolysis/dehydration of cellulose in the presence of a proper acidic catalysts.[8] These make HMF an attractive starting material for various chemical syntheses”
25) Page 9, line 325: “monomers” is plural!
26) Page 9, line 326: “is summarized” or “will be summarized” are the correct forms instead of “was summarized” (same mistake as at page 4, line 125)
27) Page 9, line 345: “four times” or “for four consecutive runs” are two correct forms
28) Page 9, line 352: “in the reductive amination” is the correct form
29) Page 9, lines 355‑365: “champion catalyst” does not sound very formal. Please reformulate in a more suitable fashion!
30) Page 10, line 360: “reductive amination” is the correct form
31) Page 10, line 372: “reductive amination” is the correct form
32) Page 10, lines 379‑380: “other different amines” is incorrect: please eliminate “other” or “different”! One of the two is superfluous
33) Page 10, line 385: “HMF could be quantitatively converted”: HMF is not the subject, but the object!
34) Page 10, line 393: “as active species”. The cative sites belong to the active catalyst, which is a species, not a site itself!
35) Page 10, line 401: “HMFAs” must be plural!
36) Page 11, line 410: “in rather mild reaction conditions” or “under rather mild reaction conditions” are the correct forms
37) Page 12‑13, Table 3: Please add the unit of measure for lifetime
38) Page 13, line 478: Please check and correct the sentence (if “high yield” is the subject, “can effetively be synthesized” is not the appropriate verbal predicate)
39) Page 13, line 488: “medicines, pesticides and cosmetics” is the correct form
40) Page 13, line 500: “has been a well‑known….for many years” or “is a well‑known….from many years” are the correct forms
41) Page 13, line 503: “to generate the corresponding imine”
42) Page 14, line 530: The verb is missing! Please correct in: “which was then hydrogenated”
43) Page 15, line 546: “polymer amines” must be corrected in “polymeric amines” or “amine‑containing polymers”
44) Page 15, line 548: “polymeric side reactions” must be corrected in “polymerization side‑reactions”
45) Page 15, line 555: “Diamines” and “monomers” must be plural
46) Page 16, line 567: “amine” must be corrected in “primary amines”
47) Page 16, line 576: “co‑loaded” is the correct form
48) Page 16, line 580: “acidic sites”: acidic is an adjective!
49) Page 16, line 586: “protonic acid sites” must be corrected in the form: “protic sites” or “Brönsted acidic sites”
50) Page 16, line 589: What does it mean: “Pd/ZrO2 nano-catalyst also reported to presented high performance”?!? Really bad english translation! Please correct after a careful english check!
51) Page 17, line 606: “proton” is a noun! The correct form is “protic polar solvent”, since “protic” is an adjective that can be properly used in this context!
52) Page 17, line 609: “solvents” is plural in both cases
53) Page 17, line 623: “providing N-phenyl-5-MeP in 63% yield” or, even better, “affording N‑phenyl‑5‑MeP in 63% yield” are the correct forms
54) Page 17, line 630: “desired” is an adjective (and is correct), “desire” is a noun (and is not correct)!
55) Page 29, lines 1101‑1102: please pay attention to de‑highlight ref.50 and to complete the reference with volume and page numbers!
56) Please uniform the style of bibliography! (e.g., give only intial page number or initial number‑final page number for ALL of the bibliographic references)
This reviewer is not going to highlight other typos and construction or grammar english mistakes occurring between observation 54) and observation 55). In any case, the authors are expected to carefully check by themselves with the help of a native english speaker before resubmitting an eventual revised version of the manuscript.
CONCEPTUAL MISTAKES AND OTHER OBSERVATIONS
Introduction
Not only biomass is the most abundant carbon‑based feedstock, but also chemicals derived from biomass, as single components, can be used in the synthesis of fine chemicals (e.g. Asian JOC 2021, 10, 3279; Green Chem. 2014, 16, 950-963; Adv. Synth. Catal. 2012, 354, 3180-3186; Curr. Opin. Green Sust. Chem. 2018, 14, 19‑25; Current Developments in Biotechnology and Bioengineering 2022, Chapter 11: Production of fine chemicals from renewable feedstocks through the engineering of artificial enzyme cascades, pages 261-279). The authors should point it out with appropriate references (the suggested ones are enough).
Page 3, lines 98‑107
While explaining the reaction mechanism of reductive amination, with particular and appropriate reference to the acidic and basic sites of an hypothetic catalytic system suitable for this kind of process, the authors forget to give some examples of catalytic systems, which would be highly beneficial for the readers to have a practical idea of a suitable catalyst for the process (possibly, with an additional figure).
Furthermore, at line 103, the authors talk about “the choice of the active metal”: does it mean that the catalyst for the reductive amination is mandatorily a metal‑based species? In the opinion of this reviewer, the authors should better clarify this point. Indeed, as a counter‑example, it is possible to mention a work by 2021 Nobel laureate Benjamin List (J. Am. Chem. Soc. 2006, 128, 13074), in which a chiral phosphoric acid (TRIP) is used as a metal‑free catalyst in an enantioselective reductive amination: in this case, the acidic proton of the phosphoric diester catalyst is the acidic site, while the oxygen atom of P=O bond is the basic site (Chem. Rev. 2014, 114, 9047). The latter example is not related to the particular case of reductive amination of biomass, nor related to reduction with molecular hydrogen, but the introduction of paragraph 2 of the manuscript appears to be rather general. Please be more precise!
Page 6, line 209
Imines and Schiff bases are synonyms. What is the difference between them implicitly intended to exhist by the authors? Later on, it is then impossible to understand the mechanism described from line 216 (what is ammonolysis??? What does it means that a Schiff base is turned into an imine?)
Page 7, line 272
Toluene is apolar but not aprotic! It is true that the pKa of the methyl group of toluene is extremely high (41 in water, 43 in DMSO), but a suitable strong base can deprotonate it! The extraction of protons from the methyl group of toluene to oxidize it up to benzoic acid is what makes this solvent less harmful than benzene (benzoic acid can be metabolized and benzoate salts have been used as food additives).
Page 8, line 279
How can an ester (ethyl acetate) protonate aniline???
Page 14, line 532
Once again, what is “ammonolysis”? And so on, what do the authors mean by “imine intermediate”? The mechanism is not well‑explained because is not clear in the terminology, in the current form of the manuscript. If a protonated imine intermediate is involved, please take into account that a protonated imine is called “iminium ion”.
Reviewer 2 Report
The authors review several methods for the preparation of amines from biomass-based furans. The review is timely and of importance. However some important information must be added before the manuscript is ready for publication:
1. When discussing reductive amination of FF with aniline, some reports concerning the preparation of furylamines from FF and directly from carbohydrates were omitted and should be discussed (i.e. Green Chem., 2018,20, 2494-2498)
2. When discussing the formation of cyclopentanone and reductive amination to cyclopentamine, a new section discussion the formation of aminocyclopentanes should be added. These are obtained in one step either from furfuryl alcohol (aza-piancatelli rearrangement) or from furfural. (i.e. D. Fisher, L. I. Palmer, J. E. Cook, J. E. Davis, J. Read de Alaniz, Tetrahedron 2014, 70, 4105–4110.; R. F. A. Gomes, L. A. S. Cavaca, J. M. Gonçalves, R. Ramos, A. F. Peixoto, B. I. Arias-Serrano, C. A. M. Afonso, ACS Sustain Chem Eng 2021, 9, 16038–16043.; K. Griffiths, C. W. D. Gallop, A. Abdul-Sada, A. Vargas, O. Navarro, G. E. Kostakis, Chemistry - A European Journal 2015, 21, 6358–6361.; R. F. A. Gomes, J. A. S. Coelho, C. A. M. Afonso, ChemSusChem 2019, 12, 420–425.; S.-W. Li, R. A. Batey, Chemical Communications 2007, 3759; A. Hiscox, K. Ribeiro, R. A. Batey, Org Lett 2018, 20, 6668–6672.; R. F. A. Gomes, N. R. Esteves, J. A. S. Coelho, C. A. M. Afonso, Journal of Organic Chemistry 2018, 83, 7509–7513.)
3. In Scheme 2 the reaction from 7a to 8a is not a formal reduction, this should be corrected.
Finally but least important, the writing should be polished (i.e. “In general, traditionally, amination of hydrocarbons” line 60; “In contrast, the amination of carbonyl and alcohol-based oxygenates generated from biomass is more straightforward, environmentally benign, economically feasibility.” From 61 to 63; “Moreover, theoretically, the only byproduct in theory is water” line 64.)
Once these corrections are done, and the relevant information added, I am pleased to accept the manuscript.
Reviewer 3 Report
Please check the PDF file.

Round 2
Reviewer 1 Report
The corrections suggested in the previous report by this referee have been correctly implemented and the reply from the authors is convincing. The manuscript is now suitable for publication.
Author Response
Thank you very much for your approval.
Reviewer 2 Report
The authors improved substantially the manuscript, however no efforts were given in adding the transformation of furfural to important diamino-cyclopentenone blocks, despite this being known under heterogeneous conditions. (Catalysts 2019, 9, 301– 312, React. Chem. Eng., 2023, 8, 482, ACS Sustain Chem Eng 2021, 9, 16038-16043) This information is highly relevant to the topic since these are very useful amines (being used in total synthesis and being biologically active) and are obtained in one or two steps directly from furfural under heterogeneous catalysis, including under flow conditions. (Angew. Chemie – Int. Ed. 2013, 52, 10862– 10866) After adding this discussion, the manuscript is ready to be published.
